# Localized cardiac small molecule trajectories and persistent chemical sequelae in experimental Chagas disease

Zongyuan Liu[1,2], Rebecca Ulrich vonBargen[2,3], April L. Kendricks[4], Kate Wheeler[5], Ana Carolina Leão[6], Krithivasan Sankaranarayanan [2,7], Danya A. Dean[1,2], Shelley S. Kane[1,2], Ekram Hossain[1,2], Jeroen Pollet [6], Maria Elena Bottazzi [6,8], Peter J. Hotez[6,8], Kathryn M. Jones [6,8] ✉ & Laura-Isobel McCall [1,2,7,9] ✉

Post-infectious conditions present major health burdens but remain poorly understood. In Chagas disease (CD), caused by *Trypanosoma cruzi* parasites, antiparasitic agents that successfully clear *T. cruzi* do not always improve clinical outcomes. In this study, we reveal differential small molecule trajectories between cardiac regions during chronic *T. cruzi* infection, matching with characteristic CD apical aneurysm sites. Incomplete, region-specific, cardiac small molecule restoration is observed in animals treated with the antiparasitic benznidazole. In contrast, superior restoration of the cardiac small molecule profile is observed for a combination treatment of reduced-dose benznidazole plus an immunotherapy, even with less parasite burden reduction. Overall, these results reveal molecular mechanisms of CD treatment based on simultaneous effects on the pathogen and on host small molecule responses, and expand our understanding of clinical treatment failure in CD. This link between infection and subsequent persistent small molecule perturbation broadens our understanding of infectious disease sequelae.

Antimicrobial treatment, including antibacterials, antivirals, antifungals, and antiparasitics, is a mainstay of therapeutic strategies against infectious agents, with treatment success often defined by metrics associated with pathogen clearance. However, as post-infectious irritable bowel syndrome, post-infectious Lyme arthritis, post-acute sequelae of non-persistent viruses, and many other conditions demonstrate, pathogen clearance does not always lead to symptomatic cure[1-3]. While the pathogenesis of long COVID is still under study, some postulated mechanisms include symptom persistence and/or emergence even after viral clearance[4]. The study of post-infectious sequelae, their mechanisms, and their treatment, has

unfortunately lagged compared to the study of acute infection pathogenesis and treatment. Such a need is particularly clear in Chagas disease (CD): clinical trial data has demonstrated that antiparasitic treatment with the drug benznidazole (BNZ) is insufficient to prevent disease progression or mortality if administered late in the disease, even in patients showing undetectable parasite burden[5]. This mechanism is distinct from parasitological treatment failure, with parasite persistence, which may result from a combination of the local nutritional environment, reduced prodrug activation via drug resistance mechanisms, and parasite dormancy[6-9]. Thus, a better understanding of the mechanisms by which BNZ fails to improve correlates

---

[1]Department of Chemistry and Biochemistry, University of Oklahoma, Norman, OK, USA. [2]Laboratories of Molecular Anthropology and Microbiome Research, University of Oklahoma, Norman, OK, USA. [3]Department of Biomedical Engineering, University of Oklahoma, Norman, OK, USA. [4]Southern Star Medical Research Institute, Houston, TX, USA. [5]Department of Biology, University of Oklahoma, Norman, OK, USA. [6]Department of Pediatrics, Baylor College of Medicine, Houston, TX, USA. [7]Department of Microbiology and Plant Biology, University of Oklahoma, Norman, OK, USA. [8]Department of Molecular Virology and Microbiology, Baylor College of Medicine, Houston, TX, USA. [9]Present address: Department of Chemistry and Biochemistry, San Diego State University, San Diego, CA, USA. ✉e-mail: kathrynj@bcm.edu; Lmccall@sdsu.edu

of disease may inform CD drug development, but also help guide the development of new tools and new interventions for other chronic, persistent, and post-infectious illnesses.

CD is caused by the intracellular protozoan parasite *Trypanosoma cruzi*[10]. Approximately 6 million people are infected with *T. cruzi*, of which over 1 million live with symptomatic chronic CD cardiomyopathy, resulting in more than 12,000 deaths per year[11]. CD is endemic in the Americas but has become a global health issue due to migration. Close to 70 million people worldwide are at risk of infection. After acute infection, patients without treatment usually progress to the chronic phase of disease. The chronic phase has four forms: an asymptomatic form with no apparent clinical symptoms, a cardiac form, a digestive form, and a cardiodigestive form. The cardiac form is a major cause of morbidity and mortality, with symptoms of dilated cardiomyopathy, apical aneurysms, congestive heart failure, arrhythmias, cardioembolism, and stroke[12].

Immunoregulatory mechanisms are important processes for the control of the immune-mediated damage observed in chronic CD[13–15]. Studies comparing *T. cruzi*-specific immune responses in asymptomatic patients to patients with cardiac disease have identified key aspects of the host immune response that correlate with disease severity. Patients without overt clinical disease exhibit a mixed $T_H1/T_H2/T_H17$ immune profile, with increased levels of the key cytokines IFNγ, IL-10, and IL-17. In contrast, patients with clinical signs of cardiomyopathy have a predominantly $T_H1$ immune profile, with increased levels of the pro-inflammatory cytokines IFNγ, IL-6, TNFα, and IL-1β and very little IL-10 and IL-17[16–20].

Building on the knowledge that a balanced immune response correlates with reduced disease symptoms, we developed a vaccine containing recombinant Tc24-C4 protein combined with a TLR4 agonist adjuvant[21,22]. The Tc24 antigen is a *T. cruzi* flagellar calcium-binding antigen that is conserved across multiple discrete typing units of *T. cruzi*, and is expressed in extracellular trypomastigote and early intracellular amastigote stages[23,24]. When used as an immunotherapy in experimentally infected mice, the vaccine increased levels of antigen-specific CD8+ cells as well as the key cytokines IFNγ, IL-10, and IL-17A[22,25,26]. Importantly, combining this vaccine with BNZ treatment in a vaccine-linked chemotherapy strategy led to significantly reduced cardiac pathology while reducing the amount of BNZ necessary for efficacy[25–27]. Together, these data suggest that through induction of a balanced $T_H1/T_H2/T_H17$ immune response, the Tc24-C4 immunotherapy can overcome some limitations of BNZ treatment alone in people with CD.

Small molecules (<1500 Da) include the intermediates in central carbon metabolism, the major building blocks of the cell (lipids, amino acids, nucleotides, etc.), specialized small molecules produced by microorganisms, diet-derived molecules, etc, as defined in the Human Metabolome Database[28]. Immune responses and small molecules are tightly linked. In CD, parallel or opposite gradients of parasite burden, small molecule alterations, and immune responses were observed in the heart[29,30]. The degree of overall cardiac small molecule alterations was correlated to the degree of cardiac inflammation, and to serum profibrotic cytokine levels. The cardiac levels of specific acylcarnitines and glycerophosphocholines were also correlated with inflammation, fibrosis, and the cytokines TGFβ, CTGF, and PDGF[31]. Immune responses are regulated by metabolism[32]: the balance between fatty acid oxidation and glycolysis controls effector T cell vs regulatory T cell formation[33], and de novo phospholipid biosynthesis is necessary for Th17 cell effector function and proliferation[34], for example. PC(O-16:0/2:0) and structurally-related Platelet-Activating Factor (PAF)-like lipids are produced in response to multiple pro-inflammatory cytokines, and promote inflammation, reactive oxygen species formation and fibrosis, all of which are key aspects of CD pathogenesis[35,36]. The pentose phosphate pathway, associated with the biosynthesis of nucleotide pentose precursors and the production of NADPH, is important for *T. cruzi* control[37]. Immune signaling can also directly reshape

metabolism: for example, stimulation with IFNγ and TNFα cytokines decreases cardiomyocyte dependency on fatty acid oxidation for basal respiration[38]. Metabolism-targeting strategies prevented acute CD mortality in mouse models, demonstrating a causal role of metabolism in disease progression[39]. However, the impact of antiparasitic treatment and immunomodulatory strategies such as the Tc24-C4 immunotherapy on local tissue small molecule responses in chronic CD and in post-treatment recovery is currently unknown.

To address this gap, we combined in this study mass spectrometry-based small molecule characterization with three-dimensional modeling ("chemical cartography"[29]) to investigate the spatial effects of infection, Tc24-C4 immunotherapy, and BNZ on the cardiac small molecule profile and chemical restoration at early vs late chronic infection timepoints and post-treatment. Our results provide insights into mechanisms of treatment success and demonstrate the key association between the cardiac small molecule profile in CD and treatment success in a mouse model, paving the way for further studies in humans. Furthermore, our results present a different paradigm of small molecule and metabolic determinants of chronic and post-infectious disease sequelae.

## Results

### Highly localized infection-induced small molecule perturbations

Given the characteristic localization of CD apical aneurysms[40,41], we assessed the impact of infection on the overall small molecule profiles between six heart segments: left atrium, right atrium, top half of the left ventricle free wall, bottom half of the left ventricle free wall, top half of the right ventricle free wall, and bottom half of the right ventricle free wall (Fig. 1a, $N = 15$ distinct mice per group, position and timepoint), at multiple chronic stage timepoints selected based on prior data on resolution of parasitemia, emergence and worsening of cardiac fibrosis in this parasite and mouse strain combination (see "Materials and methods" section; refs. 25,42,43). Strikingly, different cardiac regions demonstrated different overall responses to infection over time (Fig. 1b, c, Supplementary Fig. 8, Supplementary Tables 5-6). The impact of infection on the overall small molecule profile in the atria was minor at early chronic stage timepoints, as reflected by smaller PERMANOVA pseudo-F values, a measure of effect size (left atrium, PERMANOVA $p = 0.014$, pseudo-F statistic = 3.03; right atrium, PERMANOVA $p = 0.12$, pseudo-F statistic = 1.58), and either decreased over time (left atrium, Kruskal-Wallis (KW) with post-hoc Dunn's test, FDR-corrected, $p = 2.68e-18$ for distances between infected and uninfected samples at 50 days post-infection (DPI) vs 142 DPI) or remained overall unchanged (right atrium, KW with post-hoc Dunn's test, FDR-corrected, $p = 0.77$ for distances between infected and uninfected samples at 50 DPI vs 142 DPI). Similar to the left atria, the overall small molecule profile at the top of the left ventricle was only significantly affected by infection at early chronic timepoints (PERMANOVA $p = 0.005$ pseudo-F statistic = 3.65 at 50 DPI; $p > 0.05$ at 75 and 142 DPI). In contrast, the overall small molecule profile at the bottom of the left ventricle was significantly affected at all timepoints (PERMANOVA $p = 0.003$ pseudo-F statistic = 5.16 at 50 DPI; $p = 0.002$ pseudo-F statistic = 3.50 at 75 DPI; $p = 0.001$ pseudo-F statistic = 4.46 at 142 DPI), though the small molecule profile partially renormalized over time (KW with FDR-corrected post-hoc Dunn's test $p = 9.92e-17$ for distances between infected and uninfected samples at 50 DPI vs 142 DPI). With regards to the right ventricle, the overall small molecule profile was significantly and persistently affected at all timepoints at the top of the right ventricle (PERMANOVA $p < 0.05$, pseudo-F statistic=2.061 at 50 DPI, pseudo-F statistic = 4.61 at 75 DPI and pseudo-F statistic = 3.91 at 142 DPI), and from 75 DPI onwards at the bottom of the right ventricle (PERMANOVA $p < 0.05$ pseudo-F statistic = 4.28 at 75 DPI and pseudo-F statistic = 5.43 at 142 DPI). Strikingly, the magnitude of small molecule perturbation was highest in the bottom segments of both the left and right ventricles at our chronic 142 DPI timepoint,

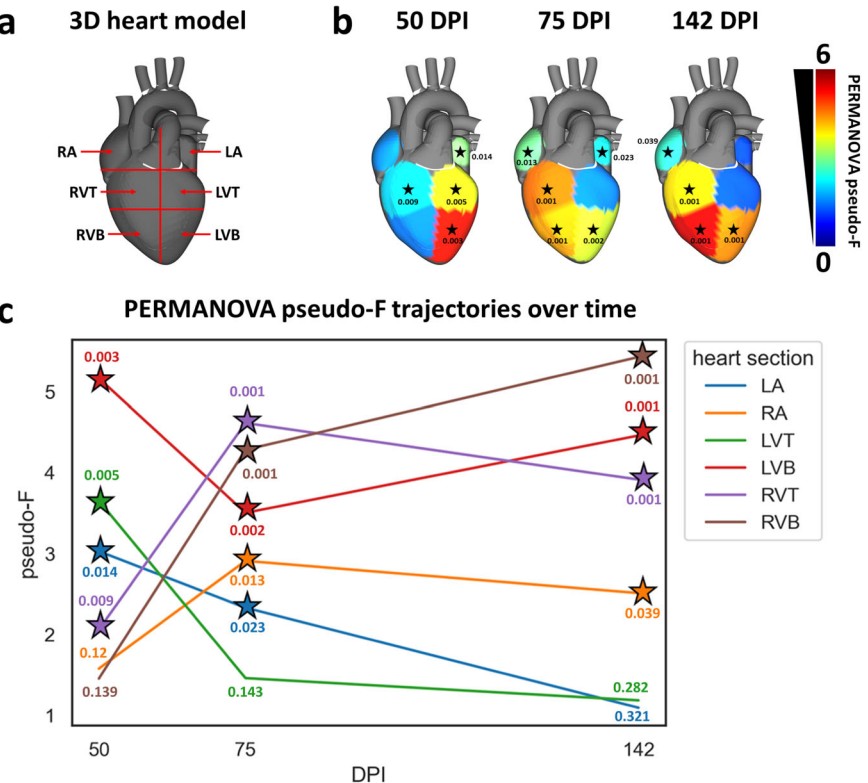

**Fig. 1 | Infection-induced small molecule perturbations are highly localized spatially and temporally. a** Heart sections analyzed. RA right atrium, LA left atrium, RVT right ventricle top, RVB right ventricle bottom, LVT left ventricle top, LVB left ventricle bottom. **b** PERMANOVA pseudo-F for PCoA distances between naïve and infected at 50, 75, and 142 days post-infection (DPI). *p-value < 0.05 by PERMANOVA. Pseudo-F is a measure of effect size: the greater the pseudo-F value, the greater the difference between infected and uninfected samples at that site and

at that timepoint. **c** Site-specific PERMANOVA pseudo-F vs time. Star, p-value < 0.05 by PERMANOVA. Pseudo-F values increase between 75 and 142 DPI for left ventricle bottom and right ventricle bottom, indicating increasing impacts of infection on the overall small molecule profile. In contrast, pseudo-F values decrease between 75 and 142 DPI for all other tissue sections, indicating decreasing differences between infected and uninfected samples at these sites and these timepoints. N = 15 mice per group and per position. Source data are provided as a Source Data file.

matching with the fact that CD patients commonly present cardiac apical aneurysms, in addition to left ventricle apical and posterior fibrosis[12,41]. This localized small molecule perturbation to the heart apex contrasts with our observation of comparable parasite load between ventricle sites at 142 DPI (KW p > 0.05, FDR-corrected), but concur with our prior observations of disconnect between parasite tropism and location of small molecule perturbation[30,39].

The specific small molecules perturbed by infection were mainly site-specific (Fig. 2a) and timepoint-specific (Fig. 2b). This indicates localized, site-specific responses to *T. cruzi*, rather than baseline differences in the small molecule profile, since there was considerable overlap across sites and timepoints in terms of small molecule absence/presence (Supplementary Fig. 1). The small molecules affected at least one timepoint represent between 0.33% of detected features (in right atrium) and 5.14% of detected features (in right ventricle top), with comparable proportions also affected in the left ventricle bottom (4.43%) and right ventricle bottom (3.46%). Strikingly, the greatest overlap between infection-perturbed small molecules across timepoints was observed in the left ventricle bottom, which may indicate a stronger connection between pathogenesis, immunity, and the small molecule profile at this site. Overall infection-perturbed small molecule features include purines, amino acids, and multiple lipids (Fig. 2c and Supplementary Data 1). Many purines were consistently depleted across sections and timepoints (*e.g. m/z* 137.046 RT 0.35 min, annotated as hypoxanthine (Fig. 2c); *m/z* 268.104 RT 0.302 min, annotated as adenosine; *m/z* 269.088 RT 0.35 min, annotated as inosine). In contrast, intermediates in their breakdown were increased, including *m/z* 169.036 RT 0.375 min, annotated as urate (Supplementary Data 1). These results are consistent with RNA-seq data in an

independent mouse cohort showing infection-associated increases in purine degradation modules (Supplementary Fig. 16). In contrast, *m/z* 307.084 RT 0.362 min (annotated as oxidized glutathione) was only significantly elevated by infection in the left ventricle bottom, whereas it was detected but not significantly elevated in the right ventricle (Supplementary Data 1).

**Impact of infection on glycerophospholipids and acylcarnitines**
Given that several of the individual infection-impacted small molecules were glycerophosphocholines (PCs), glycerophosphoethanolamines (PEs) and acylcarnitines (Supplementary Data 1), and given the association of these lipid classes with CD severity[31,39], we investigated these lipid classes in greater detail. Total PCs were significantly increased by infection in the right atrium, left ventricle bottom, right ventricle top, and right ventricle bottom at 75 DPI, whereas they were significantly decreased in the right ventricle bottom at 142 DPI (Fig. 3a). This pattern of predominant increase was also observed across mass ranges (Supplementary Figs. 17–21). Total PEs were significantly increased in the right atrium, left ventricle bottom, right ventricle top and right ventricle bottom at 75 DPI, and decreased in the left ventricle bottom at 50 DPI (Fig. 3b). This pattern was also overall consistent across mass ranges (Supplementary Figs. 28–32). Increased PCs and PEs are consistent with RNA-seq data showing elevated PE and PC biosynthesis modules (Supplementary Fig. 16 and Supplementary Table 1). Detected acylcarnitines were dominated by long-chain acylcarnitines (Supplementary Fig. 39), and were significantly increased by infection at 142 DPI in the left ventricle top, left ventricle bottom, and right ventricle top. Increased acylcarnitines is consistent with lower beta-oxidation gene expression with infection in an independent cohort and changes

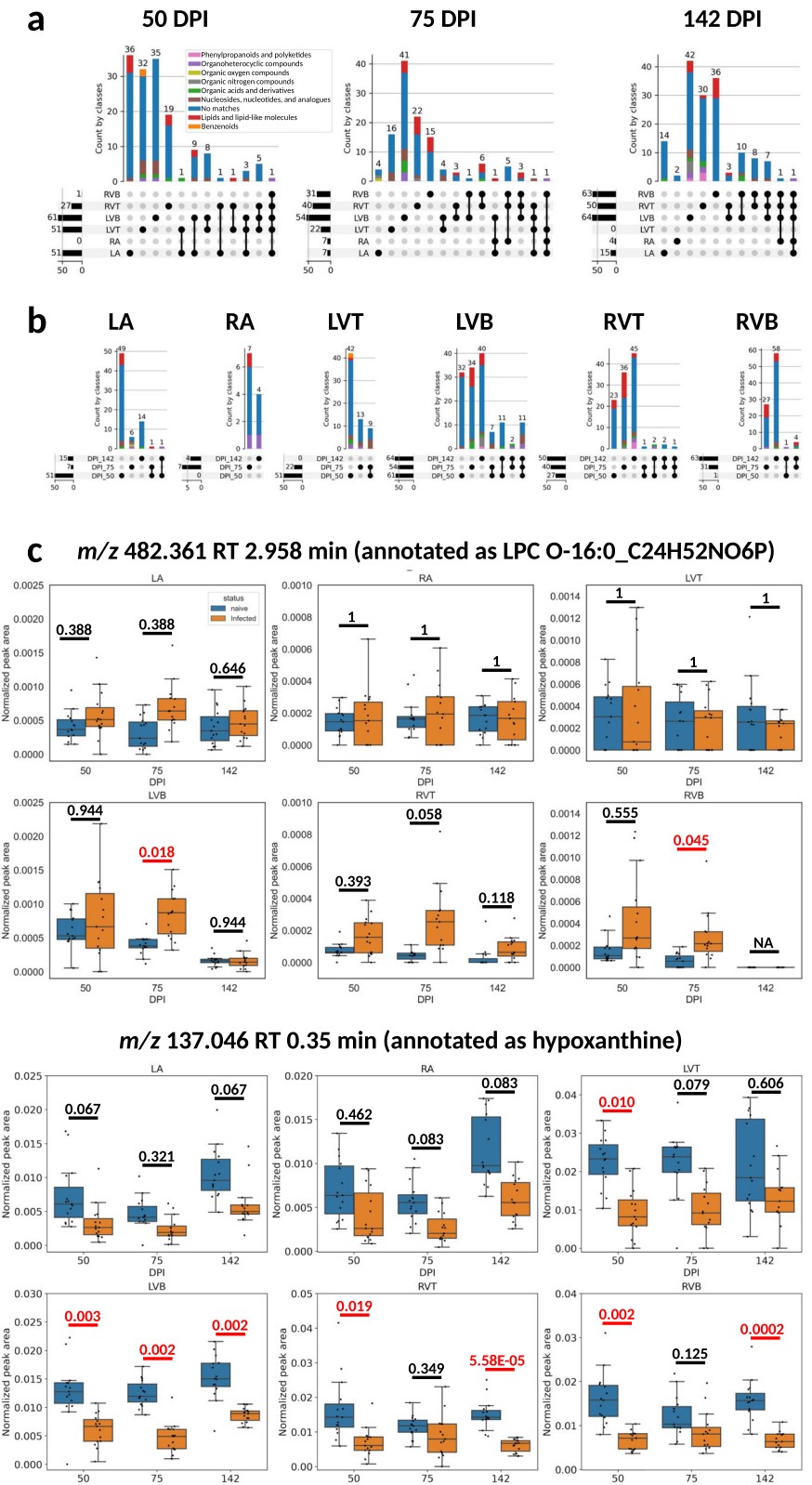

in lipid transport genes (Supplementary Fig. 16, Supplementary Data 2, and Supplementary Table 1).

**Standard-of-care BNZ does not fully restore small molecule alterations**

As the BENEFIT clinical trial demonstrated[5], BNZ treatment is insufficient to prevent adverse clinical outcomes in chronic symptomatic CD patients, including those with stage III heart failure. In contrast, greater improvement is observed if treatment is initiated in asymptomatic CD[44,45]. This lack of efficacy cannot be explained merely by parasite drug resistance, since successful parasite clearance was demonstrated by blood qPCR[5]. Persistent cardiac fibrosis is likely to contribute to this clinical failure[12], but may not be the only determinant. Based on our findings of tissue location-specific worsening small molecule impact of

**Fig. 2 | Impact of infection progress on individual small molecules. a** UpSet plot demonstrating limited overlap between infection-impacted small molecules at each heart section, for each timepoint post-infection. Bars are colored by super-class from ClassyFire annotation, as implemented in MolNetEnhancer[130,131]. Dark circles represent intersections between groups, with the size of that intersection on top of the colored bar graph. Total number of features impacted by infection at each sampling site is represented on the left barplot. **b** UpSet plot demonstrating limited overlap between infection-impacted small molecules at each timepoint, for each of the heart sections. **c** Representative small molecules locally impacted by infection over time. Red line, $p$-value < 0.05 by Mann-Whitney $U$ test, two-sided, FDR-corrected. Black line, $p$-value > 0.05 by Mann-Whitney $U$ test, two-sided, FDR-corrected. Numbers on top of the lines are the corrected $p$-values. Boxplots represent median, upper and lower quartiles, with whiskers extending to show the rest of the distribution, except for points that are determined to be outliers by being beyond the interquartile range ±1.5 times the interquartile range. RA right atrium, LA left atrium, RVT right ventricle top, RVB right ventricle bottom, LVT left ventricle top, LVB left ventricle bottom, DPI days post-infection. $N$ = 15 mice per group and per position. Source data are provided as a Source Data file.

infection over time (Fig. 1c) and our prior observations of relation between experimental CD severity and magnitude of small molecule perturbation[30,31,39], we hypothesized that an inability to restore infection-induced small molecule alterations at specific cardiac locations may contribute to the lack of clinical efficacy of BNZ in chronic symptomatic CD. To test this hypothesis, mice were treated with BNZ standard-of-care (100 mg/kg BNZ for 18 days, beginning at 69 DPI) (Fig. 4a, $N$ = 15 mice per position and per group). This timepoint was selected because parasitemia has already resolved in this mouse model (indicating transition from acute to chronic disease) and because inflammation and fibrosis are already observed[25,42]. Electrocardiographic data and echocardiographic data were acquired post-treatment, and animals were euthanized at 142 DPI. We assessed 46 phenotypic indicators of disease severity, including heart weight, liver weight, body weight, echocardiographic parameters, electrocardiographic parameters, and immunological parameters. Sixteen of these parameters showed significant differences between infected and uninfected animals (Mann-Whitney $U$ test, uncorrected, $p$ < 0.05). Only five of these parameters were restored by BNZ treatment (Table 1), even though parasite burden in BNZ-treated animals was no longer significantly different from the limit of detection at all ventricle sites ($p$ > 0.05, KW with post-hoc Dunn's test, FDR-corrected), and differed significantly from untreated animals ($p$ < 0.05, KW with post-hoc Dunn's test, FDR-corrected) (Fig. 4b, f). Immune responses, in particular, were not restored. Incomplete restoration of cardiac immune gene expression by BNZ treatment was also confirmed at the transcript level (Supplementary Figs. 14–16). Thus, our mouse model experimentation indicates only partial phenotypic efficacy of standard BNZ-treatment, modeling the clinical situation[5].

Strikingly, of the two sites most significantly impacted by infection at 142 DPI, the bottom of the left and right ventricles (Fig. 1b), BNZ only re-normalized the small molecule profile at the bottom of the right ventricle (KW with FDR-corrected post-hoc Dunn's test $p$ = 1.44e-07) (Fig. 4c, d). Nevertheless, the small molecule profile remained significantly different from uninfected animals at this site (Fig. 4c, d, PERMANOVA $p$ < 0.05), indicating that BNZ alone, followed by 56 days of recovery, is unable to fully lead to small molecule restoration. We had one outlier in the BNZ treatment group with high parasite burden in one sampling site only. However, conclusions were not affected by this outlier mouse (Supplementary Figs. 9–12 and Supplementary Tables 7–12). This persistence of small molecule alterations following BNZ treatment was confirmed in an independent *T. cruzi* infection model (Supplementary Fig. 13). Furthermore, analysis of RNA-seq metabolic modules showed that most of the differential modules between BNZ and uninfected were also differential between vehicle and uninfected, supporting the occurrence of infection-induced metabolic changes that fail to be restored by BNZ treatment (Supplementary Table 1 and Supplementary Fig. 16). In contrast, beta oxidation-associated RNA expression modules were only differential between infected untreated and uninfected groups, and not between BNZ-treated and uninfected groups (Supplementary Table 1 and Supplementary Fig. 16). This concurs with our findings of renormalization of acylcarnitines with BNZ treatment in the left ventricle top, and a pattern in the direction of restoration in the left ventricle bottom and

atria. However, no pattern of acylcarnitine re-normalization was observed in the right ventricle samples (Supplementary Figs. 40–41).

## Better small molecule restoration with combination BNZ and immunotherapy

Given the association between CD and immune responses[16] and the poor tolerability of standard BNZ treatment regimens in humans[46,47], we then assessed whether an experimental regimen consisting of a combination of reduced-dose BNZ (25 mg/kg for 18 days) followed by two doses of a therapeutic vaccine, Tc24-C4 (25 µg at 92 and 106 DPI), could provide a phenotypic and small molecule benefit. This therapeutic vaccine has previously been shown to improve cardiac fibrosis and cellular infiltration[48]. This combination treatment did not significantly reduce parasite burden ($p$ > 0.05, KW with post-hoc Dunn's test, FDR-corrected, Fig. 4b, f). However, it successfully restored CD3+CD8+IFNγ+ splenocyte cell levels that BNZ-alone treatment had failed to improve. Six disease indicators were not-restored by the combination treatment, four of which also failed to be restored by BNZ-alone treatment (heart rate, body weight, liver weight to body weight ratio and P wave amplitude, Table 1). We have previously shown that vaccine-linked chemotherapy ameliorated *T. cruzi*-induced changes in cardiac structure by approximately 30 days after treatment ended[48]. Improvements in cardiac function were previously observed by approximately 80 days after treatment end[48], indicating that the timepoints used in this study may be insufficient to observe full functional restoration after combination treatment.

The combination treatment led to improved cardiac small molecule restoration compared to BNZ alone in the right ventricle bottom (increased distance to infected samples in combination-treated animals, KW with FDR-corrected post-hoc Dunn's test $p$ = 7.43e-08) (Fig. 4c, d, Supplementary Figs. 9–12, and Supplementary Tables 7–12), and reduced the distance to uninfected samples in the right ventricle top and bottom (KW with FDR-corrected post-hoc Dunn's test $p$ = 8.018e-03 for RVT and $p$ = 3.58e-07 for RVB), though it was unable to fully restore the small molecule profile (Fig. 4c–e). In combination with our prior work[48], this suggests that metabolic restoration may precede functional restoration. These findings also indicate that parasite clearance from the heart is insufficient on its own to restore the small molecule profile to a pre-infection state, and that the inability of BNZ to fully restore the small molecule profile is not due to lingering parasite fragments.

To determine the specific small molecule pathways that fail to be restored by the different treatments or can be successfully reverted, we used machine learning (random forest[49]) approaches to first identify small molecule features that differ between uninfected and infected untreated groups at our 142 day timepoint. We then assessed which of these features were restored by treatment (see "Methods" section). Overall, we identified 27 to 64 small molecule features perturbed by infection at these cutoffs depending on the heart position. Small molecules altered by treatment were mainly site-specific, indicating local impacts of treatment (Fig. 5). These small molecule features represent 0.87% to 5.54% of the dataset, depending on the sampling site. Concurring with our observations of minimal small molecule perturbation at the left ventricle top and right atrium, almost

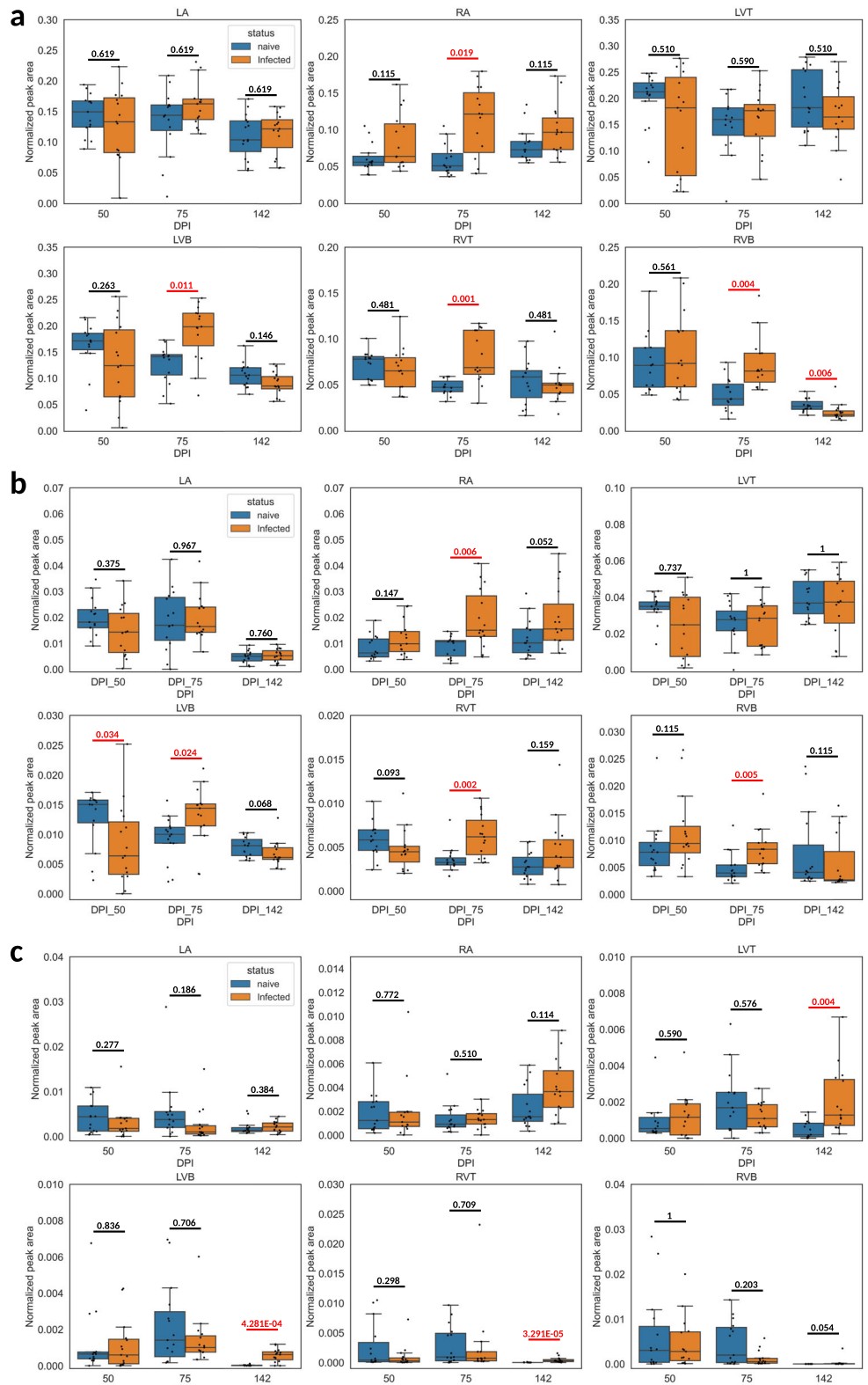

**Fig. 3 | Impact of infection duration on total glycerophosphocholines, glycerophosphoethanolamines and acylcarnitines.** Data represents summed peak areas for all glycerophosphocholines (**a**), all glycerophosphoethanolamines (**b**), and all acylcarnitines (**c**). identified as described in "Methods" section. RA right atrium, LA left atrium, RVT right ventricle top, RVB right ventricle bottom, LVT left ventricle top, LVB left ventricle bottom, DPI days post-infection. Red line, *p*-value < 0.05 by Mann-Whitney *U* test, two-sided, FDR-corrected. Black line, *p*-value > 0.05 by Mann-Whitney *U* test, two-sided, FDR-corrected. Numbers on top of the lines are the corrected *p*-values. Boxplots represent median, upper and lower quartiles, with whiskers extending to show the rest of the distribution, except for points that are determined to be outliers by being beyond the interquartile range ±1.5 times the interquartile range. *N* = 15 mice per group and per position. Source data are provided as a Source Data file.

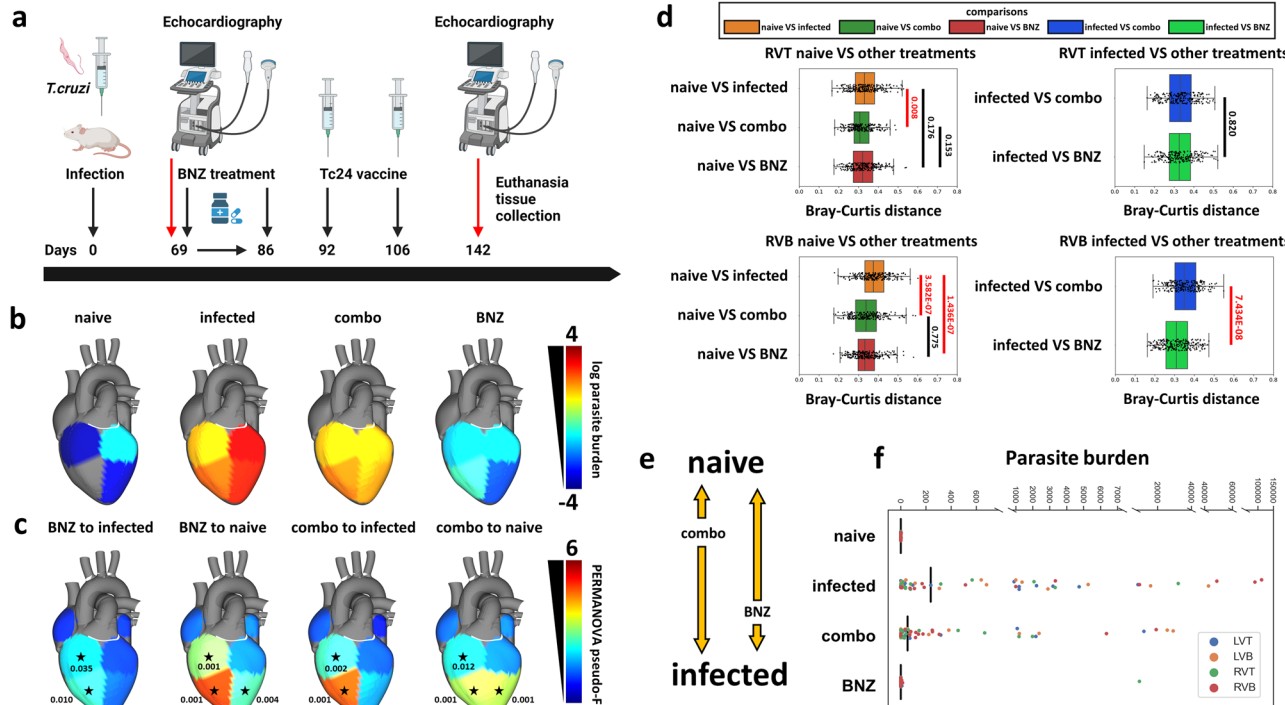

**Fig. 4 | BNZ does not fully restore small molecule alterations while vaccine-linked chemotherapy showed greater improvement. a** Treatment timeline. Created using BioRender.com. **b** Median parasite burden in each group (log scale). Right ventricle bottom values in naive mice not displayed due to log scale (median of zero). **c** PERMANOVA pseudo-F at 142 DPI for PCoA distances between naïve mice and the different treatment groups or between infected untreated mice and the different treatment groups. **d** PCoA distances between experimental groups at 142 DPI. In particular, BNZ-treated samples were not as distant from infected samples as combo-treated samples at RVB. Red lines, KW with post-hoc Dunn's test, FDR-corrected $p < 0.05$. Black lines, $p$-value $> 0.05$. Numbers on top of the bars are the corrected $p$-values. Boxplots represent median, upper and lower quartiles, with whiskers extending to show the rest of the distribution, except for points that are determined to be outliers by being beyond the interquartile range ±1.5 times the interquartile range. BNZ benznidazole, RVT right ventricle top, RVB right ventricle bottom, LVT left ventricle top, LVB left ventricle bottom. $N = 15$ mice per group and per position. **e** Diagram of restoration of small molecule status based on distance analysis from panel **d**. **f** Parasite burden in individual samples. Gaps are positioned so that no data is hidden. Source data are provided as a Source Data file.

all infection-perturbed small molecules at these sites could be restored by at least one treatment (Fig. 5b, c). In contrast, at the other heart positions, only about half of the infection-perturbed small molecules were restored by any treatment (compare Fig. 5b to Fig. 5a; Supplementary Data 3 and Supplementary Fig. 42). Several small molecules were commonly restored by both treatment regimens, likely due to the overlapping treatment composition (Fig. 5f). However, as expected based on its greater positive impact on overall small molecule profiles (Fig. 4), the combination treatment restored more small molecules than BNZ-only treatment (Fig. 5d vs Fig. 5e). For example, *m/z* 720.445 RT 3.09 min (annotated as PC 28:3;O3) in the left ventricle top and *m/z* 396.332 RT 2.991 min (no annotation) in the left ventricle bottom recovered closer to naive in mice that received the combination treatment than in mice that received BNZ only (Fig. 5g). In contrast, *m/z* 137.046 RT 0.35 min (annotated as hypoxanthine) in the right ventricle top and *m/z* 494.325 RT 2.865 min (annotated as LPC 16:1 or LPC O-16:2;O) in the left ventricle bottom recovered better in mice that received the BNZ treatment than in mice that received the combination treatment, though still incompletely.

In aggregate, some small molecule classes were more readily restored than other small molecule classes. For example, the majority of infection-perturbed lipid and lipid-like molecules were restored by treatment, except in the left ventricle bottom. In contrast, nucleosides and nucleoside analogs were predominantly not restored (Table 2). This was confirmed at the level of metabolic modules using RNA-seq analysis (Supplementary Figs. 14–16, Supplementary Data 2, and Supplementary Table 1): overlapping modules between vehicle vs uninfected and BNZ vs uninfected include multiple modules related to purine metabolism. These were increased in both comparisons,

concurring with the depleted purine levels observed in our LC-MS analysis of infected untreated samples and infected samples followed by treatment. On the other hand, beta-oxidation modules were only differential for the comparison between vehicle vs uninfected and not BNZ vs uninfected, concurring with our small molecule analysis showing easier restoration of lipids.

Given the availability of time-course tissue small molecule characterization data, we then tested the hypothesis that small molecules already perturbed at early chronic stage timepoints (50 DPI or 75 DPI) might be harder to restore by treatment. However, we did not observe a clear temporal association pattern between small molecules that were restored by treatment and those that were not (Supplementary Table 3).

### Stronger association between disease parameters and small molecules not restored by treatment

We then sought to determine whether specific small molecules that failed to be restored by treatment could be associated with persistent immune alterations or functional cardiac readouts. Given the overlap between some of these parameters (e.g. ejection fraction and stroke volume) and the high likelihood of common biological interdependence (e.g. between IL2 and IFNγ), we therefore first assessed whether each of the metadata parameters were correlated with each other. This analysis identified seven sets of strongly co-correlated or anti-correlated parameters (Supplementary Fig. 2), from which we chose one representative parameter from each category for downstream analysis. We then assessed whether restored vs not-restored small molecule features correlated with these parameters (Supplementary Table 13). Following multiple hypothesis correction, a

**Table 1 | Impact of treatments on infection-perturbed phenotypes**

| | Effect of BNZ treatment on phenotypic indicators | Effect of combo treatment on phenotypic indicators |
|---|---|---|
| CD3$^+$CD8$^+$TNFa$^+$ cells (percent of parent, from total splenocytes) | Not restored[a] | Unclear[b] |
| CD3$^+$CD8$^+$IFNγ$^+$ cells (percent of parent, from total splenocytes) | Not restored | Restored[c] |
| CD3$^+$CD8$^+$IL2$^+$ cells (percent of parent from total splenocytes) | Not restored | Unclear |
| T Wave Amplitude (volts) | Restored | Not restored |
| R Wave Amplitude (volts) | Restored | Not restored |
| P Wave Amplitude (volts) | Not restored | Not restored |
| P Duration (seconds) | Unclear | Unclear |
| QT Interval, corrected (seconds) | Unclear | Unclear |
| QT Interval (seconds) | Unclear | Unclear |
| RR Interval (seconds) | Unclear | Unclear |
| Left Ventricular Anterior Wall, systole (millimeters) | Restored | Unclear |
| Left Ventricular End Systolic Diameter (millimeters) | Restored | Unclear |
| End Systolic Volume (microliters) | Restored | Unclear |
| Liver Weight/Body Weight (ratio) | Not restored | Not restored |
| Body Weight | Not restored | Not restored |
| Heart rate | Not restored | Not restored |

See Supplementary Table 2 for median values and interquartile range.

[a] FDR-corrected $p < 0.1$ to naïve group and FDR-corrected $p > 0.1$ to infected untreated group by Kruskal–Wallis with post-hoc Dunn's test (two-sided).

[b] Not meeting criteria [a] or [c].

[c] FDR-corrected $p < 0.1$ to infected untreated group and FDR-corrected $p > 0.1$ to naïve animals by Kruskal–Wallis with post-hoc Dunn's test (two-sided).

significantly greater proportion of the not-restored small molecules were significantly correlated with these metadata parameters in the ventricle segments (Table 3, Fisher's exact test, $p < 0.05$; Fig. 6). For example, $m/z$ 608.393 RT 3.098 min, annotated as PC 22:1;O was positively correlated with IFNγ-expressing CD3$^+$CD8$^+$ cells in the left atrium. Inosine ($m/z$ 269.088 RT 0.35 min) was negatively correlated with ejection fraction in right ventricle top and bottom, and hypoxanthine ($m/z$ 137.046 RT 0.35 min) with ejection fraction and liver weight to body weight ratio in the right atrium and right ventricle top, and to liver weight to body weight ratio only, in the left ventricle bottom and right ventricle bottom. Multiple acylcarnitines were positively correlated with PR interval, liver weight to body weight ratio, or ejection fraction in the left ventricle bottom (Fig. 6).

Likewise, the maximal absolute correlation strength was overall greater for more metadata categories for the not-restored small molecule features (Supplementary Tables 18–23), indicating that our observation is not an artefact of the multiple hypothesis correction. This was particularly apparent for correlations to liver weight normalized to body weight in the left atrium, left ventricle bottom, right ventricle top, and right ventricle bottom. Additional correlations were also observed with P wave amplitude, another disease parameter that failed to be restored by either treatment (Table 1), at all sampling sites. Correlations with ejection fraction were also commonly observed, at all sites except the left atrium.

This relationship was already apparent at 75 DPI (pre-treatment) in the left ventricle bottom, where 16% of features restored by treatment were significantly correlated with metadata parameters, compared to 48% of not-restored features (Fisher's exact test, $p = 0.0198$, Supplementary Fig. 3 and Supplementary Table 14). In contrast, in the right ventricle bottom at 75 DPI, there were comparable numbers of restored and not-restored features correlated with metadata (Fisher's exact test, $p = 0.397$, Supplementary Fig. 3 and Supplementary Table 14). Example small molecules correlated with metadata at this timepoint include hypoxanthine ($m/z$ 137.046 RT 0.35 min), positively correlated with P amplitude and negatively correlated with ejection fraction and liver weight to body weight ratio in the left ventricle bottom, and several glycerophosphocholines positively correlated to

liver weight to body weight ratio in left ventricle bottom and right ventricle bottom. These observations suggest inherent biological differences between restored and not-restored small molecule features, including a tighter relationship between not-restored features and disease indicators. In contrast, there was no clear relationship between the proportion of features restored by Bz-only treatment or the combination treatment correlated to metadata across sampling sites and timepoints (Supplementary Table 14).

## Discussion

The ultimate goal of any therapeutic regimen is to return the patient to their pre-disease health state or better. In the context of infectious diseases, drug development has primarily focused on clearance of the pathogenic agent. However, there are increasingly frequent reports of conditions where this is insufficient to alleviate all patient symptoms (e.g., refs. 1–3,5). Thus, next-generation drug or immunotherapeutic development necessitates a better understanding of the recovery processes following infection and treatment. Here, we addressed these questions in the context of *T. cruzi* infection over time and following treatment with either the standard-of-care antiparasitic BNZ[12], or an experimental combination of BNZ and Tc24-C4 therapeutic vaccine, with a focus on infection-induced small molecule alterations. We elected to focus on the small molecule profile given its extensive association with cardiac function[50–52], our findings of correlations between disease severity and degree of small molecule perturbation in chronic CD[31], and our prior report of improved infection outcome via metabolic and lipidomic modulation in acute CD, independent of parasite burden[39]. Given the low parasite load during chronic *T. cruzi* infection, detected small molecules are most likely host-derived. Small molecule changes therefore reflect both direct parasite impacts on the host, as well as host compensatory mechanisms and immunometabolism[53,54].

Timecourse analysis revealed divergent magnitudes of the impact of infection on the small molecule profile over time, depending on cardiac regions (Fig. 1b, c), with the greatest small molecule perturbations in chronic-stage disease at apical cardiac segments (Fig. 1b, c), confirming in an independent infection model our prior findings[30] and

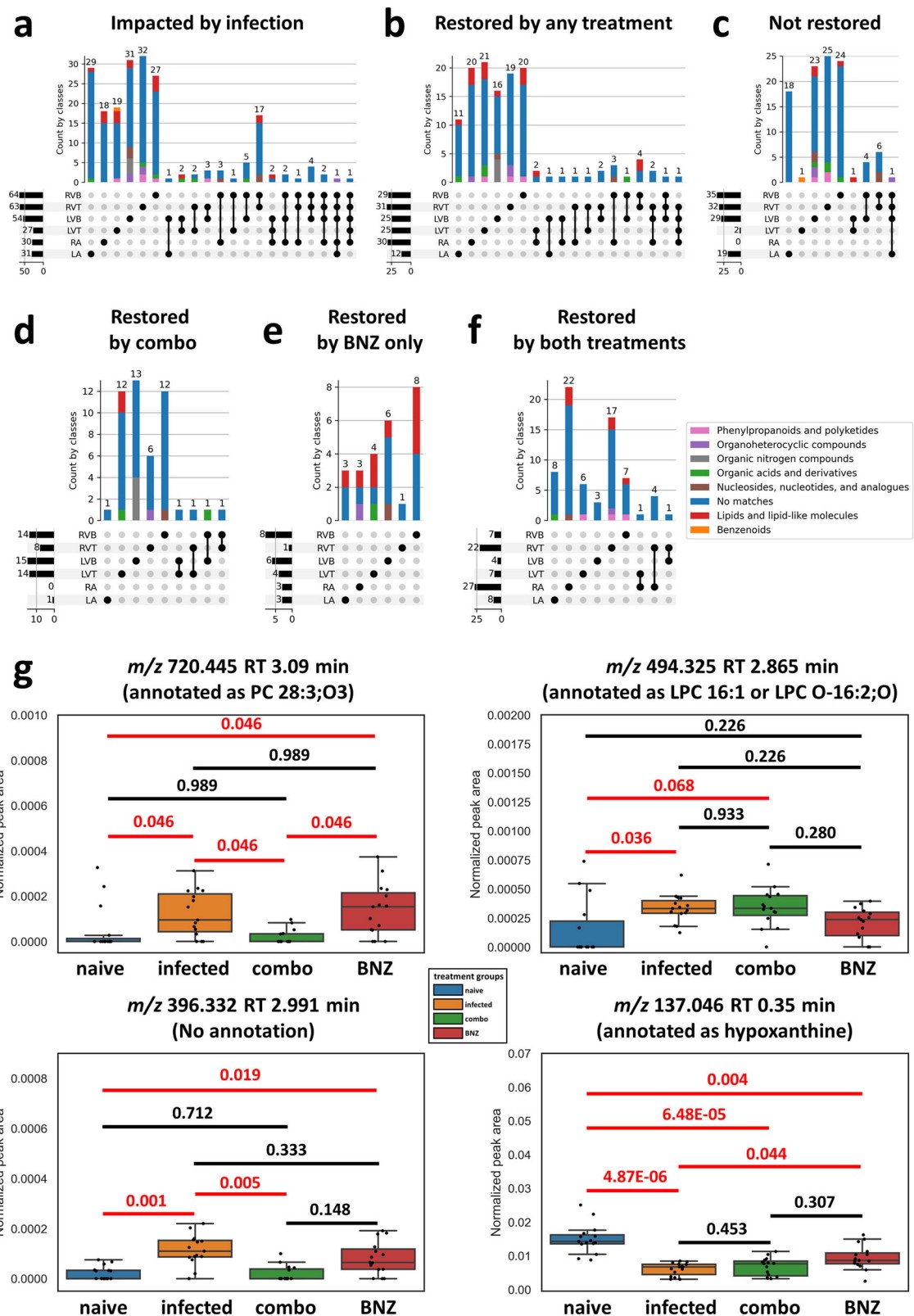

concurring with clinical observations of apical aneurysms in CD patients[12,41]. Given known differences in atrial vs ventricular myocytes[55,56] and between endocardial vs epicardial layers of the ventricle[57,58], these results may reflect inherent metabolic differences between myocytes across cardiac regions. For example, we observed here and in prior work a significant impact of *T. cruzi* infection on acylcarnitines, key intermediates in fatty acid metabolism[30,31].

Ventricular myocytes are more dependent than atrial myocytes on mitochondrial metabolism, and especially fatty acid metabolism[56], which may explain why we observed more perturbed small molecules and especially lipids in ventricular sections compared to atria. Our prior work also revealed differences in inflammation, fibrosis, and the magnitude of overall small molecule responses to infection in endocardial vs epicardial ventricle regions[31].

**Fig. 5 | BNZ + Tc24 vaccine combination treatment restored more infection-perturbed small molecules than BNZ-only treatment.** All small molecules in panel (**a**) came from random forest analysis between naïve and infected mice at 142 DPI. Bars are colored by superclass annotation generated in Classyfire[130], implemented in MolNetEnhancer[131]. **a** Small molecules impacted by infection. **b** Small molecules restored by any treatment (BNZ-only or combo treatment (BNZ+Tc24-C4/E6020-SE)). **c** Small molecules not restored by any treatment. **d** Small molecules only restored by BNZ + Tc24 vaccine treatment. **e** Small molecules only recovered by BNZ treatment. **f** Small molecules commonly restored by both treatments. RA right atrium, LA left atrium, LVT left ventricle top, LVB left ventricle bottom, RVT right ventricle top, RVB right ventricle bottom. **g** Representative small molecules restored by different treatments. Red line, *p*-value < 0.05 by Mann-Whitney *U* test, two-sided, FDR corrected. Black line, *p*-value > 0.05 by Mann–Whitney *U* test, two-sided, FDR-corrected. Numbers on top of the lines are the corrected *p*-values. Boxplots represent median, upper and lower quartiles, with whiskers extending to show the rest of the distribution, except for points that are determined to be outliers by being beyond the interquartile range ±1.5 times the interquartile range. *N* = 15 mice per group and per position throughout the figure. Source data are provided as a Source Data file.

**Table 2 | Proportion of each small molecule superclass restored by any treatment, per heart segment**

| Superclass | Post-treatment status | LA | RA | LVT | LVB | RVT | RVB |
|---|---|---|---|---|---|---|---|
| Benzenoids | Restored | /[a] | / | **0**[b] | / | / | / |
| | Not restored | / | / | 1 | / | / | / |
| Organic acids and derivatives | Restored | 1 | / | 1 | **0.5** | **0** | **0.5** |
| | Not restored | **0** | / | **0** | **0.5** | 1 | **0.5** |
| Lipids and lipid-like molecules | Restored | 1 | 1 | 0.8 | **0.25** | 1 | 0.83 |
| | Not restored | **0** | **0** | **0.2** | 0.75 | **0** | **0.17** |
| Organoheterocyclic compounds | Restored | **0** | 1 | / | **0** | 0.67 | **0** |
| | Not restored | 1 | **0** | / | 1 | 0.33 | 1 |
| Nucleosides, nucleotides, and analogs | Restored | / | 1 | / | **0.33** | **0** | **0.33** |
| | Not restored | / | **0** | / | 0.67 | 1 | 0.67 |
| Phenylpropanoids and polyketides | Restored | / | / | 1 | **0** | 0.33 | 1 |
| | Not restored | / | / | **0** | 1 | 0.67 | **0** |
| Organic nitrogen compounds | Restored | / | / | / | 1 | / | / |
| | Not restored | / | / | / | **0** | / | / |
| Organic oxygen compounds | Restored | / | / | / | / | / | / |
| | Not restored | / | / | / | / | / | / |

[a] "/" means not perturbed by infection at this site.
[b] Values ≤ 0.5 in bold.

**Table 3 | Proportion of restored and not-restored small molecule features that were correlated to disease severity metadata (142 DPI)**

| Sections | Restored by any treatment (%) | Not restored by any treatment (%) | Fisher's exact test (two-tailed) |
|---|---|---|---|
| LA | 58 | 74 | *p* = 0.447 |
| RA | 23 | All features were restored by treatment | Not applicable |
| LVT | 16 | 100 | *p* = 0.0427 |
| LVB | 4 | 72 | *p* = 0.000000168 |
| RVT | 35 | 63 | *p* = 0.0446 |
| RVB | 10 | 91 | *p* = 0.0000000000186 |

The observed disconnect between sites of small molecule alterations and sites of high parasite load (Fig. 1b vs Fig. 4b) also concurs with our prior observation in cardiac and gastrointestinal tissues[30,39] and in vitro[59], and the low parasite burden sometimes disconnected from lesion sites in human CD patients[12,60–63]. Given the association between immune alterations and the small molecule profile observed in this study and in our prior work[31], and this disconnect between parasite load and location of small molecule changes, the different small molecule alterations observed between cardiac regions may reflect local variability in immune responses, rather than direct effects of *T. cruzi*. They may also reflect prior parasite colonization, followed by immune-mediated parasite clearance but persistent immune activation at these sites.

Strikingly, we observed that, although BNZ treatment successfully cleared the cardiac parasite burden (Fig. 4b, f), this only restored half of the phenotypic indicators of disease (Table 1) and was insufficient to fully restore the small molecule profile in the left ventricle bottom, right ventricle bottom and right ventricle top (Fig. 4c, d and Supplementary Figs. 9–12, and Supplementary Tables 7–12). These results concur with our findings of persistent impairment in the urine small

molecule profile following chronic-stage BNZ treatment[64]. This expands mechanisms of treatment failure beyond the known role of cardiac fibrosis in persistently-impaired cardiac function in late-stage CD patients[12] or of parasite resistance to antiparasitics[65,66]. This hypothesis should now be tested in humans, for example using metabolomics of clinically-accessible biofluids. In contrast, BNZ shows better efficacy clinically when treatment is initiated earlier following infection[12] or in asymptomatic patients[44,45]. We likewise previously observed good small molecule restoration with acute-stage BNZ treatment[39], or in this study in the chronic stage at sites such as the right atrium and left ventricle top that were less chemically perturbed. This may reflect greater tissue small molecule resilience if the parasite is rapidly cleared or cleared prior to the establishment of broad small molecule alterations. However, at the individual small molecule level, we did not observe a clear association between duration of individual small molecule perturbation and ease of individual small molecule restoration (Supplementary Table 3).

Purines were particularly poorly restored by treatment as a class overall. Cardiac purine signaling is overall cardioprotective; in particular, adenosine can reduce cardiac hypertrophy and cardiac

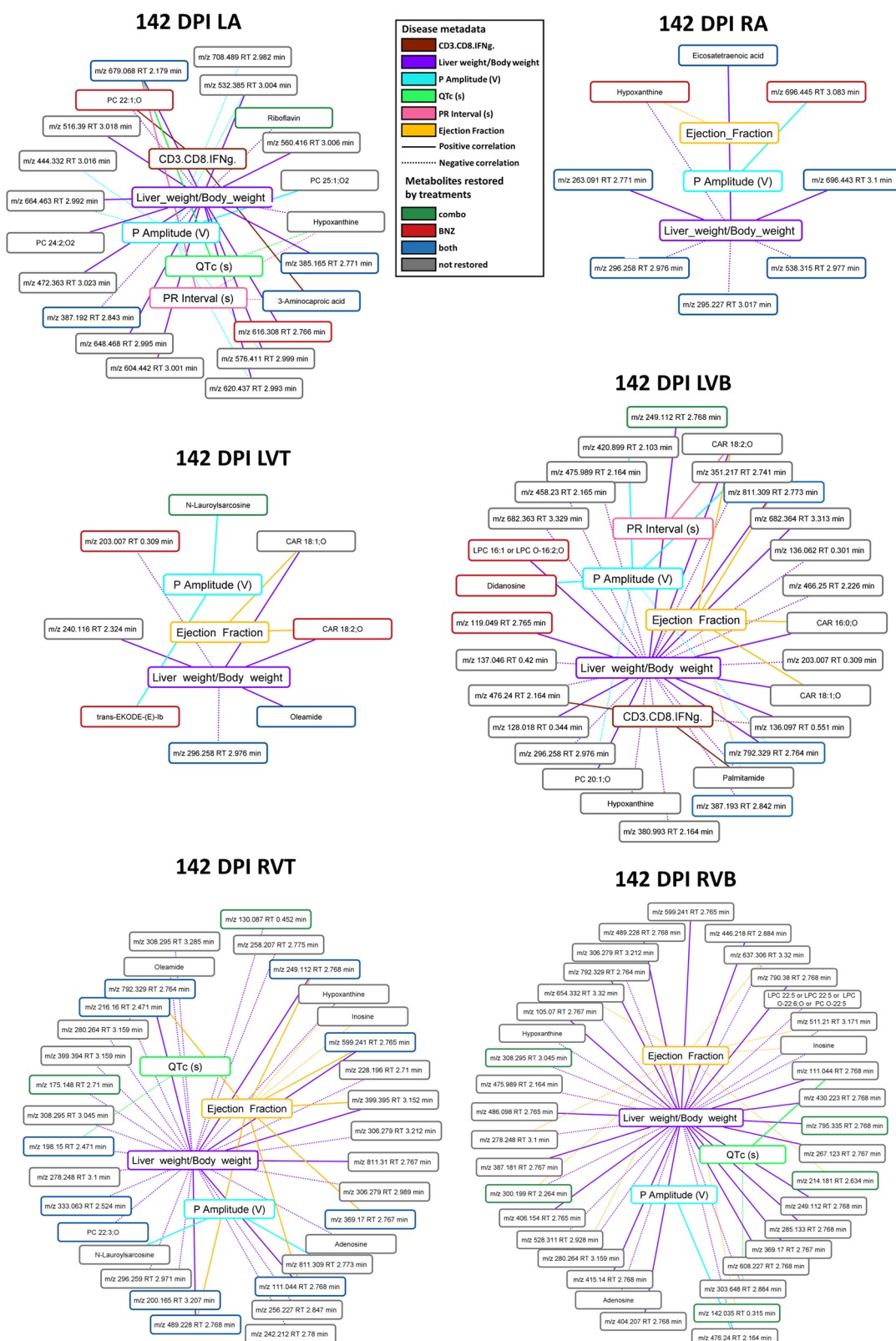

**Fig. 6 | Correlation between disease metadata and small molecule peak area at 142 DPI.** Solid lines indicate a positive correlation coefficient between disease metadata and small molecules; dotted lines indicate a negative correlation coefficient between disease metadata and small molecules. Correlation line colors relate to the specific correlated metadata category. Small molecule box colors indicate feature restoration status: green for features restored by combination treatment, red for features restored by BNZ treatment, blue for features restored by both treatments, and gray for features not restored by any treatment. Note the greater number of not-restored features correlated to the metadata. RA right atrium, LA left atrium, RVT right ventricle top, RVB right ventricle bottom, LVT left ventricle top, LVB left ventricle bottom, DPI days post-infection. N = 15 mice per group and per position. Source data are provided as a Source Data file.

fibrosis[67,68]. Adenosine is also anti-inflammatory[69,70], so persistently-depleted purine levels, as observed in this study, may contribute to the observed persistently affected TNFα levels. Thus, impaired purinergic signaling may contribute to CD progression through multiple aspects[69] (Fig. 7). Given that BNZ treatment clears parasite load, post-treatment purine depletion cannot be due to current parasite purine scavenging[71], though could reflect prior and incompletely restored parasite-mediated depletion. The gut microbiome is persistently affected by *T. cruzi* infection, even post-treatment[72–74]. Since the microbiome has been shown to regulate purine availability[75,76], persistent microbiome compositional and functional changes could be responsible for perpetuating lower levels of purines post-treatment. Indeed, a recent metagenomic analysis of fecal samples from BALB/c mice acutely infected with *T. cruzi* showed lower levels of bacterial genes associated with purine biosynthesis[73]. Treatment of CD patients with allopurinol, a purine catabolism inhibitor, failed to reduce parasite burden[77] but was associated with re-normalization of cardiac function[78], and may thus be worth re-examining in combination with existing or experimental antiparasitic agents based on our findings.

Encouragingly, combining BNZ with the Tc24-C4 therapeutic vaccine provided greater immunological and chemical benefits than BNZ-only treatment, even though parasite clearance was less than with the high-dose BNZ-only treatment (Table 1 and Fig. 4b–f). These findings concur with several prior studies reproducibly demonstrating improved treatment efficacy of vaccine-linked chemotherapy, in mouse models of acute infection with significant reductions in tissue parasite burdens as well as cardiac inflammation and fibrosis[25,26], and in chronic mouse models with improved cardiac pathology, structure,

and function[27,48,79]. Prior work has demonstrated that curative BNZ treatment increases parasite-specific memory CD8 + T cells, but at later timepoints than those used in our study: 24-48 months after treatment in humans[80,81] and 60 days after treatment in mice[82]. Additionally, Dzul-Huchim et al. showed that vaccine-linked chemotherapy using a bivalent antigen vaccine and a low dose of BNZ increased antigen-specific central memory CD8+ and CD4+ cells by 121 days after completion of treatment, but cardiac parasites were not completely eliminated[27]. In this study, we did not specifically enumerate memory T cells. Thus, we are unable to confirm the impact of vaccine-linked chemotherapy or curative BNZ treatment on the memory cell population. We were also unable to assess treatment effect on animal survival in this study: since this is not an acutely lethal model, no infection- or treatment-induced mortality were observed during the duration of the experiment, and all endpoints were well before the end of the mouse lifespan. This was a necessary design, due to the need to obtain comparable samples across all experimental groups for small molecule analysis, something that would be impossible if some animals were dying.

Our study design did not enable us to delineate which specific components of the combination treatment were responsible for this small molecule restoration effect, or whether they present with synergy. In a recently published study evaluating the impact of vaccine-linked chemotherapy on cardiac structure and function, we did show that the combination treatment significantly improved several parameters at later timepoints, including left ventricular posterior wall thickness, diameter, and volume as well as ejection fraction[48]. Thus, our results here concur with these prior findings and suggest

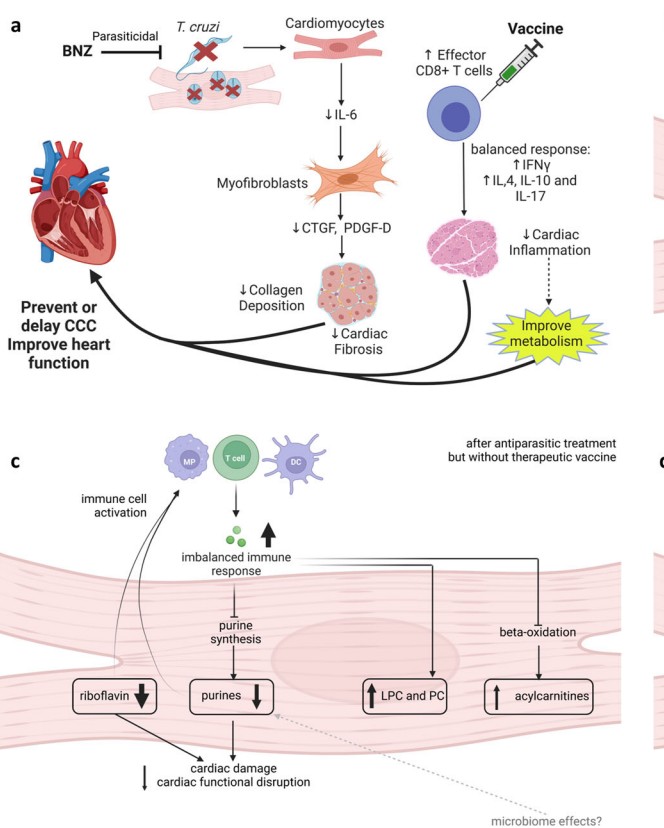
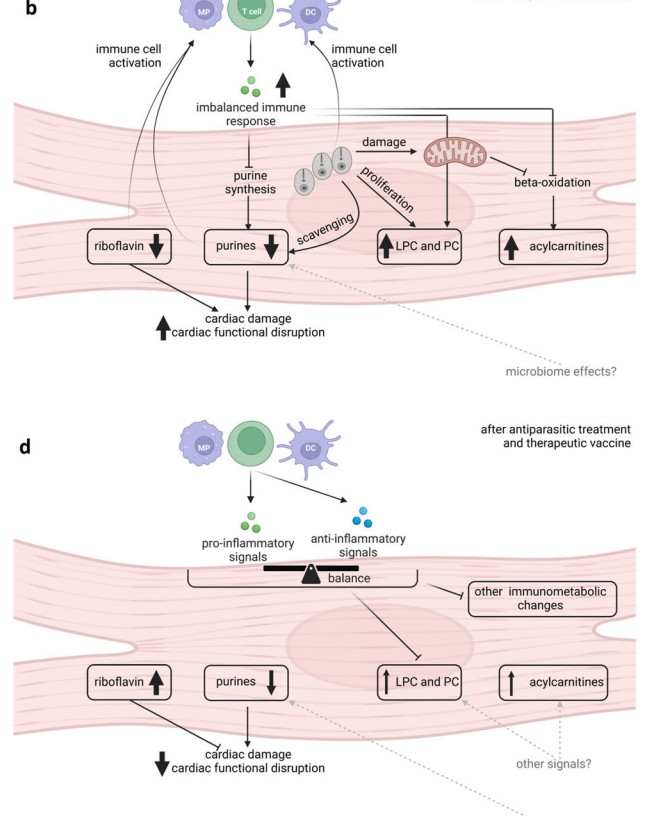

**Fig. 7 | Conceptual representation of the intersection between parasites, immune response and metabolism, in this parasite and mouse strain combination. a** Overall treatment design and expected impacts. **b** Interactions before antiparasitic treatment. **c** Interactions after antiparasitic treatment but without therapeutic vaccine. **d** Interactions after antiparasitic treatment and therapeutic vaccine. BNZ benznidazole, CCC chronic Chagasic cardiomyopathy, LPC lysophosphatidylcholines, PC phosphatidylcholines, MP macrophage, DC dendritic cell. Figure created using BioRender.com.

that improved cardiac metabolism may precede functional improvements. Low dose BNZ alone induced partial improvement, improving left ventricular diameter and volume, while neither the vaccine, containing both the Tc24-C4 protein antigen and the E6020 adjuvant, nor the E6020 adjuvant alone induced any improvements in cardiac structure or functional parameters[48]. However, vaccine alone reduced cardiac cellular infiltration and fibrosis, suggesting that reduced inflammation may contribute to metabolic restoration[48].

In combination with our timecourse findings and prior work by us and colleagues[30,39,83–85], these results demonstrate that successful CD treatment will necessitate not only parasite clearance, but also the lessening of host tissue dysfunction, and may indicate a need to redefine the CD target product profile for therapeutics and for assessment of treatment efficacy[25–27,39,86,87]. Clinical dose-reduction of BNZ, when implemented as part of a combination treatment regimen, may also help lessen treatment adverse effects[47,88]. Our approach also provides a method to test new interventions for their superiority to BNZ-only treatment prior to clinical implementation, as demonstrated here with BNZ+Tc24-C4 vaccine combination treatment.

One limitation of this study is the fact that we could not confirm sterile cure via immunosuppression, due to the need to collect cardiac tissue for chemical analysis. Thus, it is possible that the persistent small molecule alterations in the BNZ treatment group may be due to low-level parasite persistence below our detection limit or to regular cardiac re-invasion from gastrointestinal reservoirs[89], which were not analyzed here. However, a recent analysis of the urine small molecule profile in mice treated with BNZ and then immunosuppressed, demonstrated that even mice that achieved sterile cure failed to re-normalize their urine small molecule profile, further demonstrating that post-treatment small molecule alterations are not driven by parasite persistence[64]. Furthermore, parasite persistence in any tissue is unlikely to be the dominant cause of small molecule perturbations, as improved small molecule profile is observed in the BNZ+Tc24-C4 group (combo treatment group) even though parasite burden is higher in this study, and in carnitine-treated animals in acute infection even with parasite burden comparable to vehicle[39]. These results also demonstrate that persistent small molecule alterations post-BNZ treatment are not from leftover parasite fragments (unlike what is observed in post-infectious Lyme arthritis[90], for example). Instead, given that only the combination treatment and not BNZ-only treatment restored CD3⁺CD8⁺IFNγ⁺ cell levels in our study, this may reflect persistent immunological imbalance that can only be restored by an immunotherapeutic and not by BNZ-alone. Excess pro-inflammatory responses and insufficient anti-inflammatory responses can cause metabolic alterations, including mitochondrial dysfunction, switch from mitochondrial oxidation to glycolysis, reduced lipid oxidation, and reduced purine nucleotide synthesis[38,91–94]. Lysophosphatidylcholines can promote IFNγ production, and in turn, IFNγ and other inflammatory signals can promote the production of bioactive lysophosphatidylcholine and responsiveness to these signals[95–97]. Anti-inflammatory signals also shift glycerophosphocholine profiles[98]. Riboflavin, restored by combination treatment only, decreases inflammatory cytokine production and has anti-oxidant effects that could reduce tissue damage[99,100] (Fig. 7). Prior work on BNZ treatment showed improvement of infection-induced immunological perturbations[101], but with incomplete IFNγ restoration. Our own RNA-seq analyses likewise show persistent immune alterations subsequent to BNZ treatment (Supplementary Fig. 15). Indeed, three of the small molecule features that failed to be restored by treatment in the left ventricle bottom were correlated with CD3⁺CD8⁺IFNγ⁺ cell levels. Active follow-up work in our laboratory is assessing the direct relationship between IFNγ levels and CD cardiac small molecule alterations.

Animal models of CD have proven useful for identifying specific disease mechanisms and testing novel therapeutics[102]. However, all animal models to date have limitations and do not fully mimic human disease pathophysiology[103]. In this study, we used female BALB/c mice infected with a bioluminescent clone of the *T. cruzi* H1 strain. We previously demonstrated that chronic *T. cruzi* H1 infection in both inbred BALB/c and outbred ICR mice induced significant endpoint cardiac inflammation and fibrosis, similar to the pathology observed in cardiac tissue from human cases of CD[43,104–106]. We have also demonstrated that chronic *T. cruzi* H1 infection of female BALB/c mice significantly increased cardiac strain as measured on echocardiography and magnetic resonance imaging, similar to human cases of CD[43,107]. Further, in this model, we observed significant alterations of cardiac small molecules, including purines, acylcarnitines and glycerophosphocholines, that were associated with cardiac fibrosis and inflammation, similar to perturbations observed in other mouse models of chronic infection and human cases of end-stage heart failure due to CCC[30,31,108,109]. Together, these results suggest that our model replicates key aspects of the tissue pathology and clinical cardiac disease observed in human cases. However, arrhythmias and electrocardiographic abnormalities, which are frequently associated with CD in human cases[110], were not observed in this model. Further, left ventricular enlargement and significantly reduced ejection fraction on echocardiography is also a frequent finding in severe cases of CD in humans[111]. In our model, overall left ventricular enlargement is not observed and reduced ejection fraction has only been observed in a subset of mice with cardiac dysfunction[43]. Additionally, some studies have shown that male Chagas patients have a higher incidence of developing cardiac disease compared to females[112]. To date, we have only established disease progression parameters in female mice infected with the H1 strain. Therefore, future studies will incorporate both male and female mice to establish disease progression parameters in male mice, and compare clinical disease and tissue pathology between male and female mice in different stages of infection.

Our approach also does not provide cellular-level insight into hyperlocal processes, even though these are clearly also important to CD pathogenesis[113,114]. Single-cell metabolomics of in vitro *T. cruzi* infection have demonstrated bystander effects of infection on infection-adjacent but uninfected cells, as well as similar alterations to phospholipid profile compared to our chemical cartography analyses[59]. These bystander effects on non-parasite-containing cells may help explain why parasite clearance is insufficient to restore the small molecule profile to a pre-infection state.

A last, necessary limitation of our tissue analyses is that they cannot be performed sequentially on the same animals due to the invasive nature of the sample collection. Thus, each timepoint and treatment sample is derived from a different animal. While biofluids could have been sampled non-invasively, they do not necessarily reflect the small molecule changes occurring at the site of disease processes. Validation in humans would be desirable, though generating adequately controlled studies with short post-mortem interval to avoid effects on metabolism would be challenging. Our results do however concur with findings in human serum following treatment with the antiparasitic nifurtimox, which likewise showed a lack of small molecule restoration in some patients by positive mode mass spectrometry analysis up to three years post-treatment[115]. Longer-term follow-up studies should also be performed, in particular to monitor functional restoration subsequent to metabolic restoration, though these are limited by the lifespan of the mouse model.

Overall, these results have major implications for CD drug development, expanding candidate mechanisms explaining prior clinical treatment failures, as well as providing a path to assess the superiority of novel treatment regimens in pre-clinical animal models. Importantly, these results and our prior work during acute-stage treatment and in vitro infection[39,59] confirm a spatial disconnect between clinical outcomes and parasite burden that should inform CD drug development. Our method to identify local small molecule responses to

infection and treatment also suggests broad potential to study other chronic infectious diseases and post-treatment chronic sequelae.

## Methods

### Ethical approval

This research complies with all relevant ethical regulations and was approved by the University of Oklahoma Institutional Animal Care and Use Committee under assurance number R17-035 (experiments leading to Supplementary Table 1, Supplementary Figs. 13–16, and Supplementary Data 2) or by the Baylor College of Medicine Institutional Animal Care and Use Committee under assurance number D16-00475 (experiments leading to all other data). All animal studies were conducted in strict compliance with the 8th Edition of The Guide for Care and Use of Laboratory Animals[116].

### Statistics and reproducibility – study design

Animals were infected with *T. cruzi* or remained uninfected. A subset of animals were treated with BNZ or a combination of BNZ and Tc24 therapeutic vaccine. Phenotypic data was collected (weight, tissue weight, echocardiography, electrocardiography). Animals were euthanized at different timepoints, and hearts collected and used for LC-MS analysis, qPCR analysis of parasite burden or RNA-seq. Spleens were used for in vitro restimulation and cytokine production analysis.

Sex was considered in the study design. We elected to perform the study reported in the main manuscript in female mice, to correlate with prior published results in female mice[22,25,26,48]. Experiments reported in Supplementary Table 1, Supplementary Figs. 13–16, and Supplementary Data 2 only used male mice, to correlate with prior experimentation in male mice[39].

### Statistics and reproducibility – sample size

For Supplementary Fig. 13 only: No statistical method was used to predetermine sample size. Groups were constituted from the animals that survived to treatment initiation, and then to the endpoint, leading to 6 infected vehicle-treated mice, 5 infected BNZ-treated mice, and 6 uninfected mice being available for this analysis.

For Supplementary Figs. 14–16, Supplementary Table 1, and Supplementary Data 2: No statistical method was used to predetermine sample size. Groups were constituted from the animals that survived to treatment initiation, and then to the endpoint, leading to 5 mice per group being available for this analysis.

For the remainder of the manuscript: based on prior data evaluating antigen-specific immune responses after therapeutic vaccination in mice, we performed a power analysis and calculated that a group size of $n = 13$ will provide 80% power and significance level of 0.05 (Wilcoxon–Mann–Whitney test). Two mice were added to each group in the event that any animals had to be removed from the study due to reaching humane endpoints prior to scheduled study endpoints. Thus, the final group size was $n = 15$ mice per group.

### Statistics and reproducibility – data exclusions

DNA could not be extracted from the atrial homogenate due to the limited sample weight and volume. LC-MS data were filtered to obtain MS1 scans that were present in at least three samples (indicating reproducibility between animals) and that were associated with MS2 spectra (to enable annotation), with good extracted ion chromatogram peak shape. Blank removal was performed, with a minimum threefold difference between blank and samples required for a feature to be retained. For Supplementary Fig. 13 only: a single outlier was observed by PCoA in the vehicle group and was excluded. Apart from this, no samples were excluded in the manuscript. One mouse in the BNZ group was observed to have higher parasite burden in Fig. 4f. We confirmed that this outlier mouse did not impact conclusions by providing data both with and without this outlier in Supplementary Fig. 12 and Supplementary Tables 7–12.

### Statistics and reproducibility – randomization and blinding

Mice were randomly allocated to experimental groups. LC-MS/MS data acquisition was performed in random order. The investigators were not blinded to allocation during treatments. This was due to the fact that only the BNZ treatment group and the combo treatment group received BNZ, and only the combo treatment group received subcutaneous injections of the Tc24 vaccine. However, small molecule analysis and final phenotypic data analysis between experimental groups (see below) were performed by different investigators, and small molecule extraction was blinded to the treatment group.

### Statistics and reproducibility – overview of statistical methods

Shapiro-Wilk normality test indicated that mouse parameters (weight, tissue weight, echocardiography, electrocardiography, and immunological parameters, see details of data acquisition below), parasite burden, and small molecule data were not normally distributed, so univariate analysis used non-parametric two-sided Kruskal-Wallis tests with post-hoc Dunn's test, with FDR correction. Multivariate analyses were performed in QIIME[117,118] or using random forest implemented in jupyter notebooks (see details below). Pairwise correlations were calculated using pandas.DataFrame.corr in Pandas python package, with Spearman method, followed by FDR correction. Two-tailed Fisher's exact test was calculated using https://www.omnicalculator.com/statistics/fishers-exact-test.

### Statistics and reproducibility – reproducibility

The major finding of this manuscript is the fact that BNZ treatment did not fully restore the cardiac small molecule profile. Results concurred between two independent experimentation cohorts, with in vivo experimentation performed independently at the University of Oklahoma and at Baylor College of Medicine and data acquisition performed separately for both cohorts (compare Fig. 4 data to Supplementary Fig. 13). This lack of restoration of the cardiac small molecule profile also concurs with findings from the analysis of urine samples from three independent mouse cohorts, in ref. 64. Analysis of the impact of the combination treatment was performed in a single in vivo experimentation cohort.

Timecourse small molecule analysis was performed in a single in vivo experimentation cohort. However, findings with regards to the heart apex being most affected and lipids being impacted by infection concur with our prior single-timepoint analyses published in two independent studies, performed at two separate institutions (refs. 30 and 31).

Furthermore, the small molecule categories and pathways identified as impacted by infection and restored or not-restored by treatment concur with the RNA-seq data presented in Supplementary Figs. 14–16, Supplementary Data 2, and Supplementary Table 1, which came from another independent infection cohort.

### In vivo experimentation – infection

For Supplementary Figs, 13–16, Supplementary Data 2, and Supplementary Table 1: *T. cruzi* strain Sylvio X10/4 (ATCC 50800) was maintained in co-culture with C2C12 mouse myocytes (ATCC CRL-1772) in Dulbecco's Modified Eagle's Medium (DMEM) supplemented with 1X Penicillin/Streptomycin and 5% iron-supplemented calf serum (Fisher catalog number SH3007203). Media was refreshed twice weekly, with tissue culture trypomastigotes (TCT) collected weekly for passaging or for infection. For infection, culture media containing TCT was collected, parasites were pelleted by centrifugation, and then resuspended in DMEM. 5-week-old male Swiss Webster mice (Charles River) were infected by intraperitoneal injection with 500,000 Sylvio X10/4 TCT or mock-injected (uninfected controls) in a final volume of 0.1 mL.

For all other experiments: The *T. cruzi* H1 strain was transfected with pTRIX2-RE9h plasmid containing the thermostable red-shifted firefly luciferase gene PpyRE9h following published protocols[89,119,120].

Briefly, MAX Efficiency® DH5α™ Competent Cells (Invitrogen, Cat#18258-012) transfected with the pTRIX2-RE9h plasmid were inoculated into LB broth (Sigma Aldrich, cat # L7658) containing 50 μg/mL ampicillin (Sigma, Cat # A5354) at a 1:1000 ratio and grown at 37 °C, 225 rpm overnight. Plasmid DNA was isolated using a Qiagen Plasmid Midi Kit (Qiagen, Cat#12143). Plasmids were digested with AatII and AscI restriction enzymes (NEB, cat # R0117S and R0558S, respectively) for 1 hour at 37 °C, then digested plasmids were loaded onto a 0.9% agarose gel and electrophoresed at 120 V for 1 hour to separate bands. The band was excised from the gel and the linearized plasmid was extracted from the gel band using a QIAquick Gel Extraction kit (Qiagen, cat# 28704), following the manufacturer's protocol.

*T. cruzi* H1 epimastigotes were grown in LIT media supplemented with 10% FBS at 28 °C until mid log phase, then $1 \times 10^8$ epimastigotes were centrifuged at 1000×*g* for 10 min, resuspended in 100 μL Nucleofector Solution (Human T cell Nucleofector Kit, Lonza, cat# VAPA-1002), and centrifuged again at 1000×*g* for 10 min. Three hundred nanograms of linearized plasmid DNA was added to a 4 mm transfection cuvette, then epimastigotes were added and mixed. Epimastigotes were electroporated twice at 25 μF, 1500 V (3.75 kV/cm), pausing 10 seconds between each pulse, using an Amaxa Nucleofector (Lonza), then incubated on ice for 10 minutes to allow recovery. Epimastigotes were then placed in a vented 50 mL culture flask with 10 mL LIT media and incubated overnight at 28 °C. Epimastigotes were then serially diluted in LIT media with 100 μg/mL G418 disulfate salt (Sigma-Aldrich, Catalog # U0750-5G) antibiotic in a 96 well tissue culture plate and incubated at 28 °C for 42 days, refreshing media twice weekly. To confirm bioluminescence, $5 \times 10^6$ transfected epimastigotes were centrifuged at 1000×*g* for 10 min, resuspended in 1 X PBS, centrifuged again at 1000×*g* for 10 min, then diluted in 1X Cell Culture Lysis Reagent (Promega Luciferase Assay system, cat # E4030) to a final concentration of $1 \times 10^6$ epimastigotes/mL. Parasites were lysed by freezing on dry ice, then thawing before centrifuging at 4500×*g* for 2 min. Ten microliters of lysate was added into each well of an opaque 96 well plate, then 90 μL of Luciferase Assay Reagent (Promega Luciferase Assay system, cat # E4030) was added. Bioluminescence was measured in a BioTek Epoch microplate reader (Agilent Technologies). For clonal selection of bioluminescent epimastigotes, positively transfected cultures were serially diluted 1:10 in 96 well plates in LIT media until a dilution of 1: $10^{10}$ was achieved, then incubated at 28 °C for two weeks. The last dilution with visible growth at 2 weeks was used to inoculate LIT media in T25 flasks, and grown at 28 °C to expand clones.

Stationary phase transfected epimastigotes were applied to semi-confluent monolayers of C2C12 mouse myocytes (ATCC CRL-1772) in RPMI media supplemented with 5% FBS and 1X Penicillin/Streptomycin, and cultured for 24 h at 37 °C, 5% CO₂ to allow metacyclic trypomastigotes to infect cells. Media was refreshed after 24 hours and infected monolayers were cultured for 5–7 days, refreshing media every 2 days. TCT were collected from media, washed once in cRPMI and applied to fresh C2C12 semi-confluent monolayers at an MOI of 10:1, then cultured as described for 5-7 days to obtain sufficient TCT for mouse infections. Culture media containing TCT was collected, parasites were pelleted by centrifugation, washed once with sterile medical grade saline, then resuspended in sterile medical grade saline to $5 \times 10^4$ trypomastigotes per milliliter for infection.

Female BALB/c mice (Taconic Biosciences, Inc) aged 6-7 weeks old were infected by intraperitoneal injection with 5000 bioluminescent strain H1 TCT in a final volume of 0.1 mL. In this infection model, parasitemia peaks ~28 DPI, then reverts to the baseline by approximately 42 DPI, signaling the end of the acute phase and transition to the chronic stage of infection[25]. Additionally, previous studies have demonstrated that by approximately 70 DPI, significant cardiac inflammation and fibrosis are evident in this mouse and parasite strain

combination, suggestive of early chronic infection[42]. These changes are accompanied by functional and structural alterations, as observed by echocardiography[48]. We therefore initiated treatment at 69 DPI, representing early chronic infection.

In all cases, mice were monitored daily for morbidity and any mice that reached humane endpoints were euthanized. For Supplementary Figs. 13–16, Supplementary Data 2, and Supplementary Table 1: mice were maintained under a 12 h light/dark cycle, at a temperature of ca. 70-72°F (allowable range: 68–79 °F) and humidity between 30%-70%, and fed LabDiet 5053 irradiated chow, ad libitum. For all other experiments: mice were maintained under a 14 h light/ 10 h dark cycle, at a temperature between 68 and 72 °F and humidity between 30 and 70%, and fed PicoLab Select Rodent 50 IF/6F Extruded chow (Lab Supply), ad libitum.

### In vivo experimentation - treatments and endpoints

For Supplementary Figs. 13–16, Supplementary Data 2, and Supplementary Table 1 only: 10 weeks post-infection, mice were treated for 14 days with 100 mg/kg BNZ in 10% DMSO by intraperitoneal injection or left untreated. Mice were euthanized 4 weeks post-treatment (112 DPI). Hearts were collected and either placed in RNAlater (for RNA-seq) or snap-frozen in liquid nitrogen (for LC-MS).

For all other experiments: mice were randomly divided into groups of 15 mice each and treated with 100 mg/kg BNZ (BNZ-only treatment mice, MedChem express) or 25 mg/kg BNZ (combination treatment mice) suspended in 5% dimethylsufoxide/95% HPMC solution (0.5% hydroxypropylmethylcellulose, 0.4% Tween 80, 0.5% benzyl alcohol) by oral gavage once daily for 18 days. Mice in the combination treatment group then received 25 μg Tc24-C4 protein + 5 μg E6020 adjuvant in a stable squalene emulsion (Tc24-C4/E6020-SE) by subcutaneous injection twice, two weeks apart, at days 92 and 106 post-infection (Fig. 4a). Recombinant Tc24-C4 protein was expressed and purified according to previously published protocols[22], summarized as follows.

Briefly, *E. coli* BL21 (DE3) containing the pET41a (EMD Millipore) expression vector containing the Tc24-C4 sequence were inoculated into sterile LB medium (LB Difco) containing 50 μg/mL kanamycin and incubated overnight at 37 °C with 210-240 rpm agitation. A fermentation vessel containing 110 L of E. coli BSM media (5.0 g/L K₂HPO₄, 3.5 g/L KH₂PO₄, 3.5 g/L (NH₄)2HPO₄, 40.0 g/L glycerol, 4.0 mL/L 25% MgSO₄ x 7 H₂O (added after sterilization and cooling)) with 1 mL/L K-12 Bacterial Trace Salts (composed of 5.0 g/L NaCl, 1.0 g/L ZnSO₄ x 7H₂O, 4.0 g/L MnCl₂ x 4 H₂O, 4.75 g/L FeCl₃ 6H₂O,0.4 g/L CuSO₄ 5H₂O, 0.575 g/L H₃BO₃, 0.5 g/L Na₂MoO₄ x 2 H₂O, 7.5 mL/L 10 N H₂SO₄), 10 mL/L of 15 g/L CaCl₂ x 2 H₂O, 1 mL/L kanamycin (100 mg/mL) was inoculated with an aliquot of the overnight *E. coli* culture, adjusted to a starting cell density of OD₆₀₀ 0.05. When cell density reached an OD₆₀₀ 0.5, fermentation temperature was decreased to 30 °C, and when cell density reached an OD₆₀₀ 0.6-1.0 IPTG was added to a final concentration of 1 mM for induction. After 5 hours of induction, fed-batch medium (50% glycerol (v/v), 20 mM MgCl₂ was added at 3 mL/L/hr, and agitation was set at 500 rpm, 1 vvm air flow, pH 7.2, and dissolved oxygen (DO) of 30%. After 18 hours of induction, biomass was collected by centrifugation at 12,227 ×5, 4 °C, for 45 minutes, then frozen at -80 °C until purification. For purification, frozen biomass was thawed, resuspended 1:20 w/v in extraction buffer (50 mM Tris-HCl, pH 8.0), and homogenized 3-5 times at 15,000 psi on ice. Homogenate was centrifuged three times at 17,700 x g, 4 °C for 45 minutes. The supernatant was filtered first through a 0.45 μm filter, then a 0.22 μm filter before loading onto a Tricorn 10/200 column (GE Healthcare) packed with Q Sepharose XL resin (QXL, bed height 19.5 cm; column volume 15.3 mL) that had been pre-equilibrated with extraction buffer. Columns were washed with 2-3 volumes of extraction buffer followed by 8-9 volumes of Tris-HCl, pH 8.0, 75 mM NaCl. The protein was eluted with 20 volumes of 50 mM Tris HCl, pH 8.0, 125 mM NaCl, then the concentration of NaCl was adjusted to

200 mM with 5 M NaCL before passing through a stripped and re-equilibrated QXL column to remove residual endotoxin. The flow through was then concentrated using a Centricon Plus-70 (Millipore) with a 10 kDa cut off, then the concentrated protein was loaded onto a Sephacryl S-200 HR HiPrep 26/60 column (GE Healthcare) equilibrated with 1X PBS pH 7.4. Protein was eluted with 1X PBS pH 7.4 and concentration was determined spectrophotometrically using A280 measurements. Endotoxin levels were measured using the Endosafe PTS System (Charles River) and determined to be less than 20 EU/mg protein.

The TLR4 agonist adjuvant E6020 (Eisai, Inc) was formulated in a stable squalene emulsion (SE). Vaccine formulations consisting of the recombinant Tc24-C4 protein and E6020 in 100 μL of a 2% squalene emulsion in PBS 1x pH 7.4 were freshly prepared and mixed just before administration[22]. Naive age-matched mice and infected mice that were left untreated were included as controls. Mice were humanely euthanized at 50 DPI, 75 DPI (no-treatment timepoints), or 142 DPI, and hearts were excised. Fibrosis in this parasite and mouse strain combination is already observed at the first timepoint[42], worsening over time[43]. The atria were removed in their entirety, then left and right ventricles were separated and divided into top and bottom sections. Only the left ventricle free wall was collected, without the septum. All samples were snap-frozen on dry ice, then stored at -80 °C until further analysis.

## In vivo experimentation - echocardiography and electrocardiograms

Data was acquired once per animal post-treatment. Mice were anesthetized by inhalation of 2–3% isoflurane delivered by precision vaporizer to maintain an appropriate level of anesthesia for examination[121]. Fur from the ventral thorax of anesthetized mice was removed with depilatory cream, then mice were positioned in dorsal recumbency on a temperature-regulated stage set at 37 °C. Core body temperature was monitored by rectal thermometer and maintained between 36.5 °C and 37.5 °C[121]. Heart rate was monitored by Doppler electrocardiogram. Prewarmed ultrasound gel was applied to the thorax, and short-axis images of the left ventricle were obtained from the left parasternal window with a Vevo 2100 imaging system (version 1.6.0 (build 6078), FujiFilms Visualsonics, Inc.). M-mode images were obtained at the papillary level to determine left ventricular chamber dimensions and wall thickness. Immediately after echocardiographic evaluation, mice were transferred to a Rodent Surgical Monitor (Mouse Monitor, Indus Instruments) to obtain Lead II electrocardiograms. Electrocardiogram tracings were acquired on a Mouse Monitor system connected to a Power Lab System and LabChart Pro software, version 8. M-mode images were objectively analyzed using the Auto LV analysis in VevoLab software (Fujifilm Visualsonics) to measure left ventricular wall thickness and left ventricle chamber dimensions and minimize user bias. Electrocardiogram tracings were analyzed using LabChart Pro software, version 8, to measure conduction intervals and wave amplitudes.

## Splenocyte restimulation for immune evaluation

To prepare single-cell splenocyte suspensions, spleens were mechanically dissociated through 70 μm cell strainers, red blood cells were lysed with ACK lysis solution (Gibco), then washed with RPMI supplemented with 10% fetal bovine serum (FBS), 1X Pen/Strep and L-Glutamine (cRPMI). Live cells were quantified using a Cellometer Auto 2000 and AOPI live/dead dye (Nexcelom), then adjusted to a final concentration of $1\times10^7$ cells/mL in cRPMI. For each sample, $1\times10^6$ live splenocytes were restimulated for 96 hours with 100 μg/mL recombinant Tc24-C4 protein or cRPMI (unstimulated) at 37 °C, 5% $CO_2$. As a positive control, splenocytes incubated for 6 hours with 20 ng/mL phorbol myristate acetate (PMA) (Sigma-Aldrich) and 1 mg/mL ionomycin (Sigma-Aldrich) were included.

## Evaluation of cytokine-producing splenocytes

To quantify antigen-specific cytokine-producing CD4+ and CD8+ cells, cells were restimulated as described, with the addition of 4.1 μg/mL Brefeldin A (BD Biosciences GolgiPlug, Cat# 555029, Lot# 2321493) for the last 6 hours of incubation. Restimulated splenocytes were collected, washed with 1X PBS, and stained with Live/Dead fixable blue viability dye (Thermo Fisher Scientific, Cat# L23105, Lot# 1914462), 0.5 μg anti-CD3e Pacific Blue clone 145-2C11 (Biolegend, Cat#100334, Lot # B308532), 0.1 μg anti-CD4 Alexa Fluor 700 clone RM4-5 (eBioscience, Cat#56-0042-82, Lot#2218011), and 0.25 μg anti-CD8a PerCP-Cy5.5 clone 53-6.7 l (BD Bioscience, Cat#551162, Lot#5345668). Splenocytes were then fixed with Cytofix/Cytoperm (BD Biosciences) and permeabilized following manufacturer instructions. Permeabilized splenocytes were stained with 0.15 μg anti-IFNγ APC clone XMG1.2 (eBioscience, Cat#17-7311-81, Lot#4289682), 0.15 μg anti-IL-17A clone TC11-18H10.1 (Biolegend, Cat#504133, Lot# B380067), 0.3 μg anti-IL-2 Brilliant Violet 510 clone JES6-5H4 (Biolegend, Cat#503833, Lot# B320882), 0.15 μg anti-IL-4 PE-Cyanine7 clone BVD6-24G2 (eBioscience, Cat#25-7042-42, Lot#2181838), and 0.1 μg anti-TNFα PE clone MP6-XT22 (eBioscience, Cat#12-7321-82, Lot#2124591). Samples were acquired on a LSR Fortessa Cell Analyzer (BD Biosciences) using BD FACSDiva software v9.0, and 25,000 total events in a live gate were analyzed using FlowJo 10.8.1 software.

Cells were gated for total leukocytes based on forward and side scatter, then single cells based on FSC height by area, then SSC height by area. Within the single-cell population, live CD3+ cells were gated based on viability dye exclusion and positive staining for CD3 surface antigen. Within the live CD3+, cells were gated for positive staining for either CD4 or CD8 surface staining. Within either CD4+ or CD8+ populations, cells were gated for intracellular staining of IL-2, IL-4, IL-17, IFNg, or TNFa. The gating strategy is illustrated in Supplementary Fig. 4. The fluorescence minus one (FMO) strategy was used to enumerate cytokine-producing cells from splenocytes, where analysis gates were set on CD3+CD4+ or CD3+CD8+ cells stained with all cytokines (IL-4, IFN-γ, IL-2, IL-17A, and TNF-α) except for the cytokine for which the gate is being determined (Supplementary Fig. 4). To evaluate antigen-specific responses, the percent of unstimulated cells was subtracted from cells stimulated with Tc24-C4 protein for each sample, and reported as the percent of the parent population according to the gating strategy described above.

Mouse parameters (weight, tissue weight, echocardiography, electrocardiography, and immunological parameters, see below) were shown not to be normally distributed using the Shapiro-Wilk normality test, and thus were analyzed with non-parametric Kruskal-Wallis with post-hoc Dunn's test, with FDR correction for multiple comparisons between treatment groups.

## Small molecule extraction

Extraction batches included equal representation of samples across groups, with one extract per tissue segment per mouse. Extractions were batched based on tissue section and timepoint. Within a given batch, extraction was blinded to treatment group. Tissue was extracted as recommended by Want et al. for tissue small molecule characterization[122], and as in our prior work (e.g.[29,31,39]). Specifically, tissue was extracted using LC-MS-grade water (Fisher Optima) at a constant ratio of 50 mg of tissue in 8,000 μL of water, followed by 3 min homogenization at 25 Hz in a Qiagen TissueLyser with a 5-mm steel ball. A portion of the homogenate was stored at -80 °C for subsequent DNA extraction (see below). LC-MS-grade methanol (Fisher Optima) spiked with 4 μM sulfachloropyridazine was added to the rest of the homogenate to achieve a final 50% methanol concentration, then homogenized again at 25 Hz for 3 min. After the homogenate was centrifuged for 10 min at 14,980×g at 4 °C, the supernatant was dried in a Savant SPD111V (ThermoFisher Scientific) SpeedVac concentrator overnight. For organic extraction, 3:1 (v:v) dichloromethane (Fisher

Optima)−to−methanol solvent mixture spiked with 4 µM sulfa-chloropyridazine was added to the insoluble fraction from the previous step and re-homogenized at 25 Hz for 5 min. Extracts were centrifuged for 10 min at 14,980 x g at 4 °C, and air-dried overnight. Both extracts were stored at -80 °C until LC-MS analysis.

## DNA extraction

DNA extraction was performed on the water homogenate from the ventricles using the Quick-DNA Miniprep Plus Kit for solid tissues (Zymo Research), with deviations as follows: 95 µL of the water homogenate from the small molecule extraction was used during the initial protocol step; proteinase K digestion was 1 h in duration. Insufficient material remained from the small molecule extraction to enable DNA extraction for the atria.

## qPCR

qPCR was performed as previously described[29] using primers ASTCGGCTGATCGTTTTCGA and AATTCCTCCAAGCAGCGGATA to amplify T. cruzi[123] and TCCCTCTCATCAGTTCTATGGCCCA and CAGC AAGCATCTATGCACTTAGACCCC to normalize to mouse reference DNA[124]. Each DNA sample was diluted to 180 ng/µL for analysis, and analyzed in duplicate with each primer, using PowerUp SYBR Green Master Mix (Thermo Fisher). A serial dilution was performed from DNA extracted from tissues spiked with known parasite numbers, diluted with DNA from uninfected mice, to generate a standard curve. Data was acquired on a LightCycler 96 (Roche), with preincubation at 95°C for 600 s, 40 cycles of 3-step amplification (95°C for 30 s, 58°C for 60 s, 72°C for 60 s), and a melting step (95°C for 60 s, 55°C for 30 s, 95°C for 1 s). Data was processed using LightCycler 96 software, version 1.1.0.1320 (Roche), to obtain Cq values. Parasite burden in samples was determined using a standard curve, in excel software. Values below the limit of detection were marked as 0 parasite burden in the analysis. Shapiro-Wilk normality test indicated non-normal data, so non-parametric two-sided tests were used for analysis.

## LC-MS/MS data acquisition

Prior to LC-MS/MS data acquisition, dried aqueous and organic extracts were resuspended with 50% methanol as recommended by Want et al.[122] (Fisher Optima; LC-MS grade), spiked with 2 µM sulfadi-methoxine (Sigma-Aldrich) as internal standard, then both extracts were combined. Data acquisition was performed under the control of XCalibur and Tune software (ThermoScientific). Five pooled quality controls were injected at run start, to ensure system equilibration. Data was then acquired in random order for each sampling site, with a blank followed by a pooled quality control every 12 injections (Supplementary Fig. 5), building on the results by Martinez-Sena et al.[125] showing little to no impact on peak area in the second sample after a blank, and no more than 0.025 min retention time shift. We have further confirmed comparable retention times and comparable chromatographic peak areas for samples preceding vs following the blank and pooled QC (Supplementary Fig. 6 and 7). Blanks are from tubes that went through the extraction procedure but without tissue samples. Pooled quality control samples represent an equal volume pool by sampling site of all samples, generated following sample resuspension; 5 µL injection volume was used. Injection volume for each sample was 30 µL. Data acquisition was performed a single time per sample. Instrument was calibrated using Pierce LTQ Velos ESI Positive Ion Calibration Solution (Thermo Scientific). LC separation was performed on a Thermo Scientific Vanquish UHPLC system with 1.7 µm 100 Å Kinetex C8 column at 40 °C. Mobile phase A was water with 0.1% (v:v) formic acid and mobile phase B was acetonitrile with 0.1% (v:v) formic acid. The LC gradient was: 0−1 min, 2% B; 1−2.5 min, ramp up linearly to 98% B; 2.5−4.5 min, hold at 98% B; 4.5−5.5 min, ramp down to 2% B; 5.5−7.5 min hold at 2% B (Supplementary Table 4). Reversed phase separation is one of the recommended approaches for small molecule

analysis from tissues[122]. Furthermore, this separation method was similar to our previous work on CD, thus enabling cross-study comparison and the detection of chemical classes that we had observed to be perturbed by T. cruzi infection under other experimental conditions[29–31,39,126].

MS/MS analysis was performed on a Q Exactive Plus (Thermo-Scientific) mass spectrometer. Ions were generated by electrospray ionization and MS spectra acquired in positive ion mode (Supplementary Table 4). We elected to only perform positive mode data acquisition because of the large numbers of samples analyzed in this project, and because annotation rates in positive mode were three times greater than in negative mode in previous work by our group[39], likely due to the smaller numbers of negative mode spectral libraries available. Instrumental performance was assessed throughout data acquisition using a standard mix of 6 known compounds at run start, run end, and every 100 samples. This separation and MS acquisition protocol leads to a linear relationship between small molecule peak area and small molecule abundance in the sample across a broad range of molecule classes, including nucleobases, amino acid derivatives, and lipids[126].

## LC-MS/MS data processing

MS Raw data were converted to mzXML format by MSConvert software[127]. Converted mzXML files were processed in MZmine version 2.53 (see Supplementary Table 4 for parameters). Data were filtered to obtain MS1 scans that were present in at least three samples and were associated with MS2 spectra for annotation, with good extracted ion chromatogram peak shape. Blank removal was performed, with a minimum threefold difference between blank and samples required for a feature to be retained. Total ion current (TIC) normalization to a constant sum of 1 was performed in Jupyter Notebook using R version 3.6.1 (http://jupyter.org).

Principal coordinates analysis (PCoA) and PERMANOVA analyses were performed in QIIME and visualized in EMPeror[117,118,128].

Shapiro-Wilk normality test indicated non-normal data, so non-parametric two-sided tests were used for analysis. Distances were compared using non-parametric Kruskal-Wallis tests with post-hoc Dunn's test, FDR-corrected. For time series data, Mann−Whitney two-sided $U$ test with FDR correction was performed to identify features with FDR-corrected $p < 0.05$ for naïve VS infected without treatment at at least one of the 50, 75, or 142 DPI timepoints, followed by random forest analysis to rank their importance in differentiating between groups, using 1000 trees[49]. Features with Variable Importance > 2.1 were retained. The impact of treatment was likewise assessed using random forest analysis. First, a random forest classifier comparing uninfected and infected untreated groups at 142 DPI was built using 1000 trees. Features with Variable Importance > 2.1 were retained. To identify small molecules restored by the different treatment regimens, the original small molecule feature table was filtered to this subset and Mann−Whitney two-sided $U$ test with FDR correction performed between each treatment group to the naive group. Features with FDR-corrected $p > 0.05$ represent features restored by these treatments.

The overlap between features perturbed at the different timepoints or restored by the different treatments was visualized using UpSet plots[129] version 0.6 in python 3.8. The total size of each set is represented on the left barplot. Intersections are represented by the bottom plot, and their occurrence is shown on the top barplot. Barplot colors indicate small molecule superclass as determined by Classyfire[130], implemented in MolNetEnhancer[131]. Boxplots represent median, upper, and lower quartiles, with whiskers extending to show the rest of the distribution, except for points that are determined to be "outliers" by being beyond the interquartile range ±1.5 times the interquartile range. Boxplots and heatmaps were generated using Python 3.8.

Feature-based molecular networks were created using the Global Natural Products Social Molecular Networking platform (GNPS). Molecular networking groups features with similar MS2 spectra into subnetworks, enabling analysis of structurally related molecules and annotation propagation from direct library matches to related features[132,133]. The parameters to build these networks and for library searches were: 0.02 Da for both precursor ion mass tolerance and MS/MS fragment ion mass tolerance, ≥0.7 cosine score, ≥4 matched peaks, and 100 Da maximum analog search mass difference. Annotations obtained from molecular networking include direct library matches, where an MS2 spectra in the data directly matched with spectra in the GNPS spectral libraries, as well as annotations obtained through network annotation propagation, where an initial match to spectral libraries can be extended to related features (as grouped by molecular networking with the same spectral similarity cutoffs)[132,133]. All reported annotations, which were collected by an automated script from the GNPS output (https://github.com/camilgosmanov/GNPS)[134], are within 10 ppm and based on visually-inspected MS2 spectral matches, at Metabolomics Standards Initiative confidence level 2[135]. Annotations and annotation propagation were further inspected for chemical plausibility of the observed mass differences and for biological plausibility, as recommended by Theodoridis et al.[136]. Annotation rates of infection-impacted small molecules were ca. 20%, no lower than the range reported for other small molecule characterization studies outside our group (e.g.[137,138]). Overall detected chemical classes were annotated using Classyfire[130] in MolNetEnhancer[131]. Detected Classyfire[130] subclasses across all detected small molecule features included fatty acid esters, amino acids, peptides, multiple glycerophospholipid subclasses, amines, carbohydrates, fatty acids, fatty amides, alcohols, purines, and eicosanoids.

Pairwise correlation between disease parameters and restored or not-restored small molecules was calculated using pandas.DataFrame.corr in Pandas python package, with Spearman method. FDR-corrected p values were obtained using statsmodels.stats.multitest.fdrcorrection from the statsmodels 0.14.0 python package. Correlation data was visualized using Cytoscape version 3.9.1[139].

Two-tailed Fisher's exact test was calculated using https://www.omnicalculator.com/statistics/fishers-exact-test.

Glycerophosphocholines (PC), glycerophosphoethanolamines (PE), and acylcarnitines (CAR) were extracted from LC-MS/MS data using Mass Spec Query Language (MassQL), implemented in the GNPS framework[132,140]. The query parameters for acylcarnitines were based on their characteristic MS/MS fragmentation patterns[141], as follows: "QUERY scaninfo(MS2DATA) WHERE MS2PROD = 85.0285:TOLERANCEMZ = 0.01:INTENSITYVALUE > 71000 AND MS2PROD = 60.0809:TOLERANCEMZ = 0.01:INTENSITYVALUE > 21000". The query for PC was also based on diagnostic MS/MS fragmentation peaks, from the phosphocholine head group[30,142–145], as follows: "QUERY scaninfo(MS2DATA) WHERE MS2PROD = 184.0739:TOLERANCEMZ = 0.01:INTENSITYVALUE > 71000 AND MS2PROD = 125.0004:TOLERANCEMZ = 0.01:INTENSITYVALUE > 21000 AND MS2PROD = 104.1075:TOLERANCEMZ = 0.01:INTENSITYVALUE > 21000 AND MS2PROD = 86.09697:TOLERANCEMZ = 0.01:INTENSITYVALUE > 220000". The query for PE was based on the characteristic neutral loss of the head group in positive mode[142], as follows: "QUERY scaninfo(MS2DATA) WHERE MS2NL = 141.02". The output from MassQL was then filtered by superclass information from Classyfire[130] to make sure the structures output were lipid-like molecules. The resulting $m/z$ were then filtered based on inclusion in the LIPID MAPS Structure Database (LMSD and COMPDB)[146,147] in the appropriate structural classes, with $[M+H]^+$ as expected adduct and a mass tolerance of +/- 0.01 $m/z$. The final output features are the confirmed lipids molecules, which were then analyzed for impact of infection duration and treatment.

Figures 4a and 7 were created with BioRender.com. Spatial mapping on the heart model was performed with 'ili[148] on a 3D heart model purchased from 3DCADBrowser.com (http://www.3dcadbrowser.com/).

## Reporting summary

Further information on research design is available in the Nature Portfolio Reporting Summary linked to this article.

## Data availability

The metabolomics data generated in this study have been deposited in the MassIVE database under accession codes MSV000092090 [https://doi.org/10.25345/C5C824Q8C] and MSV000087427 [https://doi.org/10.25345/C5T23S]. The RNA-seq data generated in this study have been deposited in SRA, bioproject accession PRJNA670449. Molecular networks can be accessed at https://gnps.ucsd.edu/ProteoSAFe/status.jsp?task=dd1ef14c8a964bfd8843da96aa957d89 (feature-based molecular network) and https://gnps.ucsd.edu/ProteoSAFe/status.jsp?task=3fefbc8549604d34954aba2a95ec79df (MolNetEnhancer[131]). MassQL for lipids analysis can be accessed at https://gnps.ucsd.edu/ProteoSAFe/status.jsp?task=40c907b8ef5940ce9f81b096828543ab for PC, https://gnps.ucsd.edu/ProteoSAFe/status.jsp?task=ba03560c4a1f4e4d87aca8f45cab0cf1 for PE, https://gnps.ucsd.edu/ProteoSAFe/status.jsp?task=9e2a979e50cc4f38845b57998ece5513 for CAR. The LIPID MAPS database used to confirm lipid annotations can be accessed at: https://www.lipidmaps.org/[146,147]. The data generated in this study are provided in the Source Data file and at https://github.com/zyliu-OU/03172021-source-data (https://doi.org/10.5281/zenodo.8396895)[149]. Source data are provided with this paper.

## Code availability

Representative code has been deposited in GitHub: https://github.com/zyliu-OU/McCall-Lab/tree/main/03172021. Annotations from GNPS were retrieved using code deposited at: https://github.com/camilgosmanov/GNPS. Code and source data needed to reproduce manuscript figures have been deposited in GitHub at: https://github.com/zyliu-OU/03172021-source-data (https://doi.org/10.5281/zenodo.8396895)[149].

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

## Acknowledgements
This project was partially funded by the Southern Star Medical Research Institute and NIH award number R01AI168038 (to KMJ). Laura-Isobel McCall, Ph.D. holds an Investigators in the Pathogenesis of Infectious Disease Award from the Burroughs Wellcome Fund. This project was also supported by the Cytometry and Cell Sorting Core at Baylor College of Medicine with funding from the NIH (NIAID P30AI036211, NCI P30CA125123, and NCRR S10RR024574) and the assistance of Joel M. Sederstrom and Brandon Saxton. This project was supported by the Mouse Phenotyping Core at Baylor College of Medicine with funding from the NIH (UM1HG006348 and R01DK114356). The TLR4 agonist used in this study, E6020, was provided by Eisai, Inc with the assistance of Dr. Fabian Gusovsky. The content is solely the responsibility of the authors and does not necessarily represent the official views of the funders.

## Author contributions
L.I.M., K.M.J., M.E.B., and P.J.H. designed the project. K.M.J., A.L.K., D.A.D., S.S.K., and A.C.L. performed in vivo infections, treatments, cardiac evaluation, sample collection, and immune evaluation. J.P. provided reagents and vaccine formulations. Z.L., R.U., and E.H. performed small molecule extractions and LC-MS instrumental analysis under the supervision of L.I.M.; K.W., R.U., D.A.D., and S.S.K. performed DNA/RNA extraction and qPCR under the supervision of L.I.M.; K.S. performed the RNA-seq data analysis, visualization, and data curation. Z.L., K.W., K.S., K.M.J., and L.I.M. analyzed data. L.I.M., Z.L., and K.M.J. wrote the paper (first draft). All authors edited and contributed to the final draft.

## Competing interests
K.M.J., A.L.K., A.C.L., J.P., M.E.B., and P.J.H. currently are involved in a Chagas vaccine development program. M.E.B. and P.J.H. are listed among the inventors on a Chagas disease vaccine patent, submitted by Baylor College of Medicine. The remaining authors declare no competing interests.
