## [Peer review file · Nature Communications]

REVIEWER COMMENTS

Reviewer #1 (Remarks to the Author):

This manuscript states by the first time a relationship between metabolic alteration over time of infection and the development of cardiac manifestations in a mice model for Chagas disease. In addition, the authors explore the possibilities of metabolism restoration to pre-disease conditions by treating the infected mice with benznidazole, the most used drug against *Trypanosoma cruzi* infection, alone or in combination with a proposed therapeutic vaccine. The manuscript is original and brings to the community a very interesting and massive quantity of data that allow to put into a new context the search for new therapies against the disease. In general, the work is very well designed, organised, and written. I have only some minor (but in my opinion important) issues that are pointed out below. The major one, is that a more comprehensive information should be given to the readers on the limitations of the mouse model of cardiac Chagas disease. This will allow to better contextualize the findings with respect to the human disease. I have actually a short number of comments on some parts of the text that, in my opinion, should be reviewed:

Line 60: For this statement it is more appropriate to cite the original work that stated that (Chagas, 1908).

Line 62: As the authors are aware, these numbers are dynamic and change from year to year, depending on many updates. The citation refers to a review published 10 years ago, and therefore, it is likely based on old data. It would be more appropriate to cite a more recent study.

Line 62: This is a common statement, however, many pieces of evidence point to the existence of autochthonous infection cycles in the southern USA, for example. It could be argued that this happens in very specific spots. However, the same could be said for the epidemiological situation of some Latin American countries, such as Uruguay, for example. In the opinion of this reviewer, to point out Latin America instead of the Americas in general as the endemic region is not precise and contributes to the stigmatization of Latin American populations. I suggest changing "Latin America" to "America" or "most of the American continent."

Lines 109 – 113: Please provide more details regarding the conditions of host-cell infections required to obtain TCTs. If a specific published protocol was followed, please provide a reference to it.

Lines 125-126: Please provide a reference.

Line 133: According to the experience of this reviewer, all the timepoints mentioned in mice correspond to the chronic phase of the infection. If this is still the case in the current model, please state it. Later, in the results section (line 298), the authors refer to samples obtained at 50 days post-infection as "early timepoints post-infection," which is imprecise. It should be referred to as "early chronic infection timepoints" or something similar.

Lines 295-297: What are the indicators of clinical CD pathology in Figures 1B and 1C for the analyzed hearts? If these indicators are not shown, this looks as an overstatement. Please clarify this point.

Lines 328-331: The authors refer to "infection-perturbated metabolites". Did the authors consider that this greatest overlap could be due to local differences in inflammation? Authors should comment on this.

Lines 369-370: Please refer to my previous comment regarding a similar expression. Are the differences observed due to the infection itself or to differences in the inflammation process among different parts of the hearts?

Line 432: I think that "to improve metabolism" is not the most appropriate expression. Please consider "to restore metabolism to the pre-infection state" or something similar.

Lines 433-434: I do not agree with this statement, since it is not provided a mechanism formally linking the metabolic changes and the late-stage CD treatment. These results provide new paths toward the investigation of such mechanism, but they do not provide such a mechanism.

Lines 580-582: "1). Jointly, these findings provide a mechanistic metabolic model to explain the failure of BNZ treatment to improve cardiac outcomes in late-stage CD patients in the BENEFIT clinical trial": for the reasons explained earlier, I disagree with this statement. The current manuscript is undoubtedly, a great contribution in this direction, but I don't see in it a mechanistic metabolic model to explain BNZ failure to prevent cardiac outcomes of CD in patients. Mainly because the authors show a correlation between metabolic alterations and cardiac outcomes in a MODEL of CD, but it is not a mechanistic link between metabolic alterations and the cardiopathy.

Lines 582-583: Currently, it is assumed that BNZ also performs reasonably well in chronic non-symptomatic patients (40-50% patients cured, in most of cases preventing clinical outcomes). Therefore, maybe this statement should refer to symptomatic vs asymptomatic disease rather to earlier (vs. late?) infection.

I suggest to the authors to go through the annotated manuscript to look for a couple of typos I noted there and I did not include in this text.

Reviewer #2 (Remarks to the Author):

This is an interesting paper that uses an experimental Chagas disease model to study cardiac metabolism changes induced by disease and how treatment restores these changes. They present the effects of an immunomodulatory treatment.

Critiques:

- 1) Limitation of the chronic infected model. Authors decided to include mice at 69 days post infection. However other studies of chronic models used animals after 120 dpi (10.3389/fimmu.2021.712034) or 90 dpi (10.1093/jac/dkaa101). So what were the criteria that defined that at 69 dpi the animals in this study had already reached a chronic phase? Was it possible that these mice were still at a subacute phase?
- 2) Chronic Chagas heart disease in humans is characterized by a persistent fibrosing myocarditis. In humans, the amount of fibrosis changes over time in patients with chronic Chagas disease (10.36660/abc.20200597) Therefore, the presence of fibrosis is a very important characteristic that must be identified in chronic models of the disease. How fibrosis was quantified in this study? Were there evolutive changes in the amount of fibrosis over time before and after treatment? Were these changes different between study arms?
- 3) Authors decided to use a T. cruzi H1 strain in their study. However, one of the reasons for treatment failure in patients with Chagas disease is the infection by T. cruzi strain naturally resistant to benznidazole (10.1016/0035-9203(87)90020-4; 10.1111/tmi.12014). As the authors claim to study reasons for treatment failure, why the authors did not use the Colombian strain that is resistant to BZN (10.1128/AAC.00401-18)?
- 4) Authors use the results of the BENEFIT trial to affirm that benznidazole treatment is insufficient to prevent disease progression or mortality even in patients showing undetectable parasite burden. However, the design and data interpretation of this trial were criticized by their own authors (10.1590/0074-02760160334.). Furthermore, other papers showed the benefits of benznidazole treatment in decreasing Chagas disease progression (10.7326/0003-4819-144-10-200605160-00006; 10.1016/j.eclinm.2020.100694) and clinical events (10.1016/j.eclinm.2020.100694). Moreover, a negative T cruzi PCR in the blood samples do not guarantee that all intracellular parasites were

eliminated.

5) The main reason for failure of benznidazole treatment to change survival in patients with late stage chronic Chagas disease cardiac form is the extensive areas of transmural scar and LV remodeling. Viable myocardium tissue is replaced for scar creating reentrant circuits for life threatening arrhythmias and inducing LV remodeling with overt heart failure. After this stage, the elimination of the parasite will not restore the heart to its normal size and function. The authors did not discuss these aspects among the reasons for treatment failure in patients with chronic Chagas disease cardiac form.

6) Please correct: "CD is endemic in Latin America but has become a global health issue due to immigration" for "CD is endemic in Latin America but has become a global health issue due to migration".

7) Please, the criteria for the indeterminate form diagnosis is a normal ECG and normal x ray contrast studies of the esophagus and large bowel. Patients with the indeterminate or at the initial stages of the cardiac form can be both asymptomatic. Please, correct your sentence: "The chronic phase has four forms: the indeterminate form with no apparent clinical symptoms, ..."

8) Please, do not use the expression "Chagasic patients". The association of the people with Chagas disease (Home EN – FINDECHAGAS) does not call themselves this way. This expression should be avoided.

9) The study collaborators handling mice treatment could not be blinded. However, all other experiments could have been done by collaborators blinded to the treatment arm the animals were included. Please, further clarify blinding in this study and discuss this issue in limitations in case blinding was not enough.

10) The use of isoflurane to anesthetize mice cause cardiac depression with lower hear rate and LV fractional shortening. However, tables (1, 2 and 3) provided by the authors do not display any results. Only color codes. It is important to display the mean values of the studied parameters for the readers identify their changes with treatment.

11) Authors decided to use parametric or non-parametric tests based of their assumption of data normality. However, there are tests that should be used to assess the normal distribution of the data. Thus, please apply the appropriate tests.

12) Different cardiac regions have differences in the myocyte structure and myofibers types. Please discuss these differences in the light of your results.

13) Heart septum was included in the left ventricle specimens?

14) "We then sought to determine whether specific metabolites that failed to be restored by treatment could be associated with persistent disease symptoms." The authors did not test disease symptoms. They tested associations with findings of the echocardiogram and ECG of their mice model. Please, rephrase.

15) Discussion: "...and the low parasite burden sometimes disconnected from lesion sites in human CD patients 56,63-6". This can be discussed. The finding of few parasites colonies in a sample tissue of a patient with chronic infection does not mean that during the long term course of the disease, the parasite burden was always low in the current damaged tissue.

16) Conclusions: " Overall, these results have major implications for CD drug development, providing a mechanism to explain prior clinical treatment failures as well as a path to assess the superiority of novel treatment regimens in pre-clinical animal models. Importantly, these results and our prior work during acute-stage treatment and in vitro infection 33,62 confirm a spatial disconnect between clinical outcomes and parasite burden that should inform CD drug development." The data obtained from a clinical model in small mammals is useful to generate hypothesis for the pathophysiology of human Chagas disease. There is a long distance between an infection model and patients with decades lasting chronic Chagas disease. Therefore, the authors conclusion that their findings explain the failure of clinical treatment in humans should be toned down. The same apply for the Abstract.

17) There is no survival data comparing the two arms treatment. The main goal to evaluate the success of any treatment in patients with chronic Chagas disease cardiac form is not parasite clearance, but improved survival and quality of life.

Reviewer #3 (Remarks to the Author):

This manuscript follows on from some great previous work that has brought attention to the importance of spatial variability in metabolic dysregulation within organs affected by *T. cruzi* infection. Here the authors have applied their approach to study the impact of the front line anti-parasitic drug benznidazole (BNZ) and an experimental low dose BNZ + therapeutic vaccination combination treatment. The design of the experiment is relatively robust, using parameters for infection, treatment and downstream analysis that are comparable with common *in vivo* models used in this field, however, there are some questions about controls and replication.

The authors identify correlations that reveal relationships between pathogen load, immune responses, cardiac function and metabolism. The results show this is surprisingly complex, for example, cure of infection does not necessarily lead to full normalization of heart function and metabolism. In fact, on some measures the mice do 'better' even though the treatment was inferior in terms of parasite suppression. The data on the variability between the heart's anatomical sub-regions and on the correlation between overall metabolic perturbation and disease severity metadata are both interesting and quite convincing.

The focus on post-treatment functional outcomes and placing them in the context of immunological and metabolic restoration, rather than just efficacy of drugs against pathogens, is very welcome, having the potential to advance the field in a new direction. The findings could find clinical relevance in terms of defining candidate components of chronic Chagas heart disease that are reversible and those which are not.

The main limitations are:

1. Approach to assessing impact of treatment on infection. This is limited to a parasite-specific qPCR assay to detect *T. cruzi* DNA in heart tissue. While the assay is highly sensitive, it is not very informative about whether complete clearance of the infection has been achieved. This is because *T. cruzi* typically infects several other tissues besides the heart. This raises the possibility that systemic inflammation driven by infection of sites outside the heart is influencing the data, undermining the claim of relevance to broader "post-infectious" conditions. It might be feasible to better infer the likelihood of complete clearance if the sensitivity of the H1 strain to benznidazole (e.g. the IC50) were known to be similar to other strains where the same regimen has been proved to be curative.

2. In my view, the authors often exaggerate the similarity of their model to the clinical situation, especially the BENEFIT clinical trial outcomes. Chagas heart disease pathology is multi-faceted and highly variable in its presentation between individuals. However, the authors over-emphasize one specific aspect, ventricular apical aneurysms, which fits their data of stronger metabolic perturbation localizing to the lower parts of the ventricles. BENEFIT recruited patients with advanced heart disease and the failure of benznidazole to outperform placebo (after 5 years follow-up) was attributed to pathology having reached an irreversible stage. It seems doubtful that treating mice at 69 days post-infection with a 55 day follow-up is comparable to the extent that is implied. Of note, the authors data (Table 1) shows that BNZ treatment reverses most parameters of cardiac dysfunction, which is not concordant with the BENEFIT trial outcome. The data definitely have potential translational value for human disease, perhaps closer to the early chronic asymptomatic/indeterminate stage, but to claim the study provides a mechanism that explains the failure of BNZ in advanced cardiomyopathy patients appears unjustified.

Specific points:

3. Lines 36, 99, 433, 643 claim that the data provide a mechanism to explain failures of benznidazole in clinical trials. As set out above, I think this goes substantially beyond the limits of what the data actually show.

4. Line 63, due to immigration – suggest “due to migration”, one region’s immigrant is another’s emigrant after all.
5. Methods - Amounts of antibodies used in the flow cytometry should be stated.
6. Methods - Details of controls used to set gating coordinates for intracellular cytokine staining should be provided.
7. Methods – The experimental design means it is not possible to differentiate the effect of the parasite antigen and the adjuvant (TLR4 stimulation) component of the vaccine. It’s also not possible to know whether the low dose BNZ alone or the therapeutic vaccine alone would have reproduced the observed differences in immune, metabolic and cardiac function parameters.
8. Methods - The data appear to come from a single experiment, raising questions about reproducibility.
9. Line 291-297: “Given the specific localization of CD lesions...” and “metabolic trajectories which correlated with sites of clinical CD pathology” This needs further explanation – what do the authors mean by specific localization – to the heart per se, to the subregions, do lesions refer to infected areas, muscle tissue remodelling, aneurysms, sites of nervous system damage etc? Also, perhaps “corresponds with sites” would be more accurate than correlated with sites, since there is no quantification of pathology?
10. Figure 1C, is there a way for the p values quoted in the main text (from line 299 onward) to be shown on this figure?
11. Supplementary File 1, figure legends need more detail – what are the data points, metabolites?
12. Line 317: “Strikingly, the magnitude of metabolic perturbation was highest in the bottom segments of both the left and right ventricles ... matching with the sites of CD damage in patients” Similar to above, what type of damage are they referring to? Apical aneurysm? The cited review does not support the idea that Chagas heart disease pathology is localized only to these sub-regions of the ventricles e.g. there can be diffuse inflammation, microvascular damage and denervation in atrial regions.
13. Line 378, “even though parasite burden in BNZ-treated animals was no longer significantly different from uninfected animals at all ventricle sites”. I’m not sure making this statistical comparison is valid – uninfected animals by definition have zero parasite burden. Better to say something like the amount of parasite DNA was not significantly higher than the detection limit/threshold? Moreover, the data can’t distinguish between an animal that is completely cured of infection vs one where the amount of parasite DNA in the tissue sample is below the detection limit of the qPCR assay or where the animal is infected but there happened to be no parasites (or parasite DNA) in the piece of tissue used for DNA extraction. There can be relatively high numbers of parasites in a range of tissues e.g. skeletal muscle, gut, even when infection is not detectable in the heart. For this reason, it is important to clarify that parasite clearance (Line 431) only means clearance from the heart. This is acknowledged in the discussion (section at line 608), but what is not considered is the potential systemic effect on the heart of active infection elsewhere in the body, for example, circulating pro-inflammatory cytokines like TNF-alpha.
14. How bad is the heart disease in this model? Table 1 includes parameters that are restored or not restored, but it is hard to tell if this is recovery from a severe disease state or a mild one. Showing the data comparing these parameters for naïve vs infected untreated groups would help the reader to gauge this. It would also improve the ability to compare the model with human disease states.

15. Table 1 – The tissue corresponding to the T cell subsets should be stated – spleen? The data are referred to as %, but this means a % of what? It is unclear whether these data reflect an absolute difference in numbers of each CD8 T cell phenotype or if the %s are the result of shifts in the proportions of cells with other phenotypes.
16. Also for Table 1, what is the justification for using p-value cut offs of 0.1 rather than standard 0.05?
17. It is interesting that the combo treatment restored the splenic CD8 T cell compartment % to a naïve-like level, but parasite DNA was still present and all cardiac functional parameters were either not restored or scored as unclear impact. Conversely, the full BNZ regimen failed to normalize the CD8 T cells but most cardiac functional parameters were restored. Could this partially be explained by detection of a memory T cell response in the spleen after full BNZ treatment, a response that is absent from the combo regimen group?
18. Figure 3F – it looks like some data points have been cut off as a result of segmenting the data axis
19. Line 460, Table 2 title, “restored by treatment” – which treatment?
20. Line 497 to 525, I think this section would benefit from some specific example metabolites that come out of the analysis as candidates for being linked to persistence of heart disease symptoms. Perhaps those with the highest correlation co-efficients?
21. I find it hard to get a sense of how large the overall metabolic dysregulation is relative to the total metabolome. For example, in Figure 4 the charts encompass up to ~250 metabolites (the large majority of which are classified as “no matches”) but what kind of fraction is 250 of the total number of metabolites that were not significantly different between any of the groups?
22. Line 590, purinergic neurotransmission in the heart may be relevant to the discussion of purines.

Reviewer #4 (Remarks to the Author):

I read with interest this article on the treatment of a neglected tropical disease (Chagas) caused by *T. cruzi* in an animal model. The authors applied a typical antiparasitic (BNZ) and tested whether an immunomodulator (Tc24-C4), would ameliorate the long term cardiac effects of the disease. This is an important question and new drug treatments are vitally needed in order to combat what is likely to be an increasingly prevalent disease. While I have some considerable misgivings regarding the data collection methodology used, the biology appears to bear out that the combination therapy has a significant impact on long term disease prognosis.

The authors justify a metabolomics approach to this study by stating that there is a correlation of metabolite levels to inflammation and cytokine levels, and that these are therefore causally linked. Since this is the entire premise of the paper, a stronger link should be more clearly delineated, since it is not very obvious how fibrosis and overall heart inflammation is linked to metabolomics, especially the panel of metabolites detected in this study.

I have some concerns about the metabolomics - while sample preparation from tissue was managed very well, the sample preparation is clearly geared towards polar and non-polar extractions. These fractions are then resuspended using the same solvent and injected on a C8 column. There are several missing details - what was the flow rate of the column? This is important as injecting samples in a high strength (50% MeOH) buffer can have impacts on the retention time of some compounds. I am also somewhat concerned about the breadth of compounds resuspended from each fraction in the chosen

solvent. The combination of a weak reversed phase column and the analytical methodologies chosen have led to what I would describe as a small number (84) of annotated metabolites, nearly all of which are lipids (73). It might be better to describe the study as a lipidomics one, rather than a metabolomics one. I was also curious about the choice of running only in positive mode. Succinic acid and itaconate, two metabolites associated with inflammation, for example, are detected primarily in negative ionisation mode, and would recommend the choice of a HILIC column or GC-MS rather than a C8 column, where all the polar metabolites are very likely to be found in the wash-through. Was there a reason for the choice of methodology?

Additionally, were system equilibration runs performed prior to the batch? System equilibration runs are best practice for untargeted metabolomics, stabilising the system for the rest of the batch and the equilibration is reset by running interspersed blanks, as described in the methods, I am therefore confused as to why this was done. At the bare minimum, a PCA should be included in the supplementary data, allowing the reader (and reviewer) to assess the data quality, clustering of samples, and the reproducibility of the pooled samples.

I struggled to understand the PERMANOVA/pseudo-F trajectories figure (1 C). What do the trajectories actually show and what are the criteria for change? I would expect to see more mechanistics overall in the discussion - a figure explaining how this all works would be very helpful - why and how are these metabolites specifically changing, and how is this linked to the intrinsic biology that is going on in the tissue? There are several examples of individual metabolites changing, but little (apart from 'positive' or 'negative' correlation) linking them with the process of inflammation and fibrosis that presumably the immunomodulator is intended to remediate.

Finally, normalisation is a knotty problem in tissue metabolomics, especially instances like Chagas where fibrosis changes the composition of the tissues. In the methods section, normalisation is performed at the sample level via wet weight (50mg), and then by TIC - what was the impact of TIC normalisation, and was this a straight numerical TIC modifier, or was something more complex like quantile normalisation used?

In all, while the principles of the work are very good, the metabolomics data is not ideal to support the conclusions.

Response to reviewer comments:

Comments from Reviewer #1:

This manuscript states by the first time a relationship between metabolic alteration over time of infection and the development of cardiac manifestations in a mice model for Chagas disease. In addition, the authors explore the possibilities of metabolism restoration to pre-disease conditions by treating the infected mice with benznidazole, the most used drug against *Trypanosoma cruzi* infection, alone or in combination with a proposed therapeutic vaccine. The manuscript is original and brings to the community is a very interesting and massive quantity of data that allow to put into a new context the search for new therapies against the disease. In general, the work is very well designed, organised, and written. I have only some minor (but in my opinion important) issues that are pointed out below. The major one, is that a more comprehensive information should be given to the readers on the limitations of the mouse model of cardiac Chagas disease. This will allow to better contextualize the findings with respect to the human disease. I have actually a short number of comments on some parts of the text that, in my opinion, should be reviewed:

Response: We thank the reviewer for their appreciation of our manuscript. We have expanded our discussion of the limitations of the mouse model of Chagas disease (lines 903-928 in clean version; 976-1001 in track-changes version), and have made all the other requested modifications (details below).

1) Line 60: For this statement it is more appropriate to cite the original work that stated that (Chagas, 1908).

Response: We thank the reviewer for this suggestion and have made the requested change (line 65 in clean version (line 71 in track-changes version)).

2) Line 62: As the authors are aware, these numbers are dynamic and change from year to year, depending on many updates. The citation refers to a review published 10 years ago, and therefore, it is likely based on old data. It would be more appropriate to cite a more recent study.

Response: We thank the reviewer for this suggestion and have made the requested change (line 67 in clean manuscript version (lines 74 in track-changes version)).

3) Line 62: This is a common statement, however, many pieces of evidence point to the existence of autochthonous infection cycles in the southern USA, for example. It could be argued that this happens in very specific spots. However, the same could be said for the epidemiological situation of some Latin American countries, such as Uruguay, for example. In the opinion of this reviewer, to point out Latin America instead of the Americas in general as the endemic region is not precise and contributes to the stigmatization of Latin American populations. I suggest changing "Latin America" to "America" or "most of the American continent."

Response: We thank the reviewer for this suggestion and have made the requested change (line 68 in clean manuscript version (line 74 in track-changes version)).

4) Lines 109 – 113: Please provide more details regarding the conditions of host-cell infections required to obtain TCTs. If a specific published protocol was followed, please provide a reference to it.

Response: We thank the reviewer for this suggestion and have expanded this section of the Methods (lines 136-145 in clean manuscript version; lines 145-155 in expanded track-changes version).

5) Lines 125-126: Please provide a reference.

Response: We have added a reference as requested (line 165 in clean manuscript version (line 175 in track-changes version)).

6) Line 133: According to the experience of this reviewer, all the timepoints mentioned in mice correspond to the chronic phase of the infection. If this is still the case in the current model, please state it. Later, in the results section (line 298), the authors refer to samples obtained at 50 days post-infection as "early timepoints post-infection," which is imprecise. It should be referred to as "early chronic infection timepoints" or something similar.

Response: We agree with the reviewer that all timepoints are indeed in the chronic stage of infection and now specify it as recommended in the abstract, introduction and results section (lines 33, 121, 396 and 401 in clean manuscript (lines 36, 131, 413 and 419 in track-changes version)). We have also corrected "early timepoints post-infection" to "early chronic infection timepoints", as recommended (line 406 in clean manuscript version (425 in track-changes version)). Furthermore, we expanded our rationale for the selected timepoints in the Methods section (lines 148-156 in clean manuscript (lines 159-166 in track-changes version)).

7) Lines 295-297: What are the indicators of clinical CD pathology in Figures 1B and 1C for the analyzed hearts? If these indicators are not shown, this looks as an overstatement. Please clarify this point.

Response: We apologize for our lack of clarity. We had meant to refer to sites of apical aneurysms, which are commonly observed in Chagas disease patients. We have removed this statement (sentence now reads "Strikingly, different cardiac regions demonstrated different overall responses to infection over time.", lines 403-404 in clean manuscript, lines 421-423 in track-changes version). We then clarify at lines 427-430 (lines 447-450 in track-changes version) that we are matching our results to sites of apical aneurysm in patients ("Strikingly, the magnitude of small molecule perturbation was highest in the bottom segments of both the left and right ventricles at our chronic 142 days post-infection timepoint, matching with the fact that CD patients commonly present cardiac apical aneurysms, in addition to left ventricle apical and posterior fibrosis").

8) Lines 328-331: The authors refer to "infection-perturbed metabolites". Did the authors consider that this greatest overlap could be due to local differences in inflammation? Authors should comment on this.

Response: We agree with the reviewer that indeed common immune mechanisms may explain this metabolic overlap. Indeed, the greater metabolic restoration in combo-treated animals compared to BNZ-treated animals indicates a strong driving effect of immunity on metabolism. We have rephrased this sentence accordingly, to read: "Strikingly, the greatest overlap between infection-perturbed small molecules across timepoints was observed in the left ventricle bottom, which may indicate a stronger connection between pathogenesis, immunity and the small molecule profile at this site." (lines 459-461 in clean version; lines 497-499 in track-changes version).

9) Lines 369-370: Please refer to my previous comment regarding a similar expression. Are the differences observed due to the infection itself or to differences in the inflammation process among different parts of the hearts?

Response: We agree with the reviewer and have modified our discussion of these findings to clearly indicate a possible role for inflammation (lines 885-891 in the clean version (lines 957-963 in track-changes version)).

10) Line 432: I think that "to improve metabolism" is not the most appropriate expression. Please consider "to restore metabolism to the pre-infection state" or something similar.

Response: We have made the suggested change (line 614 in the clean version (line 662-663 in track-changes version)).

11) Lines 433-434: I do not agree with this statement, since it is not provided a mechanism formally linking the metabolic changes and the late-stage CD treatment. These results provide new paths toward the investigation of such mechanism, but they do not provide such a mechanism.

Response: We have removed this statement.

12) Lines 580-582: "1). Jointly, these findings provide a mechanistic metabolic model to explain the failure of BNZ treatment to improve cardiac outcomes in late-stage CD patients in the BENEFIT clinical trial": for the reasons explained earlier, I disagree with this statement. The current manuscript is undoubtedly, a great contribution in this direction, but I don't see in it a mechanistic metabolic model to explain BNZ failure to prevent cardiac outcomes of CD in patients. Mainly because the authors show a correlation between metabolic alterations and cardiac outcomes in a MODEL of CD, but it is not a mechanistic link between metabolic alterations and the cardiopathy.

Response: We thank the reviewer for this suggestion and have toned down our statement, which now reads: "This represents a new candidate mechanism of treatment failure, expanding beyond the known role of cardiac fibrosis in persistently-impaired cardiac function in late-stage

CD patients¹² or of parasite resistance to antiparasitics^{99,100}. This hypothesis should now be tested in humans, for example using metabolomics of clinically-accessible biofluids. ” (lines 798-802 in the clean manuscript version (lines 864-868 in track-changes version)).

13) Lines 582-583: Currently, it is assumed that BNZ also performs reasonably well in chronic non-symptomatic patients (40-50% patients cured, in most of cases preventing clinical outcomes). Therefore, maybe this statement should refer to symptomatic vs asymptomatic disease rather to earlier (vs. late?) infection.

Response: We thank the reviewer for this suggestion and have rephrased this section accordingly, as well as all other instances of this phrase. In particular, we clarify that the greater efficacy of benznidazole in early stage disease in patients is also mirrored by better metabolic restoration in animal models (lines 802-807 in the clean manuscript version (lines 868-875 in track-changes version)).

14) I suggest to the authors to go through the annotated manuscript to look for a couple of typos I noted there and I did not include in this text.

Response: We have endeavored to fix all typos.

Comments from Reviewer #2:

This is an interesting paper that uses an experimental Chagas disease model to study cardiac metabolism changes induced by disease and how treatment restores these changes. They present the effects of an immunomodulatory treatment.

Response: We thank the reviewer for their appreciation of our manuscript and for their feedback. We have made all requested modifications (see details below).

Critiques:

1) Limitation of the chronic infected model. Authors decided to include mice at 69 days post infection. However other studies of chronic models used animals after 120 dpi (10.3389/fimmu.2021.712034) or 90 dpi (10.1093/jac/dkaa101). So what were the criteria that defined that at 69 dpi the animals in this study had already reached a chronic phase? Was it possible that these mice were still at a subacute phase?

Response: In our infection model, parasitemia peaks approximately 28 days post infection, then reverts to the baseline by approximately 42 days of infection (PMC5865041). Resolution of parasitemia is a standard cutoff commonly used to indicate the transition to a chronic infection (e.g. PMC5196238, PMC7318042, PMID: 20399979; see also comment #6 by Reviewer 1, agreeing with the authors that this is the chronic stage of infection). Additionally, previous studies have demonstrated that by approximately 70 days post infection, significant cardiac inflammation and fibrosis are evident, suggestive of early chronic infection (PMC343959). We therefore initiated treatment at 69 days post-infection, representing early chronic infection. Other authors have likewise used similar timepoints to study chronic infection in mouse models (e.g.

PMC7561097). Furthermore, endpoint sampling and the final timecourse sample was collected at 142 days post-infection, later than the examples provided by the reviewer. Thus, these animals are indeed in the chronic stage of infection. We have added this information into the Methods and Results sections, lines 148-156, 401-403 and 526-529 in the clean manuscript version (lines 159-166, 419-421 and 570-573 in the track-changes version).

2) Chronic Chagas heart disease in humans is characterized by a persistent fibrosing myocarditis. In humans, the amount of fibrosis changes over time in patients with chronic Chagas disease (10.36660/abc.20200597) Therefore, the presence of fibrosis is a very important characteristic that must be identified in chronic models of the disease. How fibrosis was quantified in this study? Were there evolutive changes in the amount of fibrosis over time before and after treatment? Were these changes different between study arms?

Response: We agree with the reviewer that fibrosis is a major consideration. We have already established that this specific parasite and mouse strain combination establishes fibrosis at timepoints pre-treatment (PMC343959), with fibrosis progressing as time advances to timepoints corresponding to the treatment duration and endpoint (PMC6915297). We have also previously shown that our therapeutic vaccine reduces cardiac fibrosis (PMC9947347). We have now added this information to the methods and results section (lines 151-153, 174-175, 401-403 and 528-529 in the clean version (lines 161-164, 184-186, 419-421 and 570-573 in the track-changes version)).

3) Authors decided to use a T. cruzi H1 strain in their study. However, one of the reasons for treatment failure in patients with Chagas disease is the infection by T. cruzi strain naturally resistant to benznidazole (10.1016/0035-9203(87)90020-4; 10.1111/tmi.12014). As the authors claim to study reasons for treatment failure, why the authors did not use the Colombian strain that is resistant to BZN (10.1128/AAC.00401-18)?

Response: We agree with the reviewer that drug resistance in the parasite can be one cause of treatment failure. However, in this manuscript, we were specifically interested in the less-understood situation where parasites are successfully cleared, and yet patient symptoms are not improved. Thus, it was critical for us not to use a benznidazole-resistant strain. We apologize for our lack of clarity with regards to this issue, and have added additional explanation in the abstract (line 31 (line 34 in the track-changes version)), introduction (lines 56-62 in the clean version (lines 61-67 in the track-changes version)) and discussion (lines 798-800 (864-867 in the track-changes version)), adding the references recommended by the reviewer as #99 and 100.

4) Authors use the results of the BENEFIT trial to affirm that benznidazole treatment is insufficient to prevent disease progression or mortality even in patients showing undetectable parasite burden. However, the design and data interpretation of this trial were criticized by their own authors (10.1590/0074-02760160334.). Furthermore, other papers showed the benefits of benznidazole treatment in decreasing Chagas disease progression (10.7326/0003-4819-144-10-200605160-00006; 10.1016/j.eclinm.2020.100694) and clinical events

(10.1016/j.eclinm.2020.100694). Moreover, a negative T cruzi PCR in the blood samples do not guarantee that all intracellular parasites were eliminated.

Response: We acknowledge the limitations of the BENEFIT trial and added some of the references recommended by the reviewer (as #79 and #80). We have tempered our statements accordingly, in particular to reflect the fact that benznidazole is expected to show greater benefit in asymptomatic people and by clearly specifying that this lack of improvement is in symptomatic patients (as raised in the reference mentioned by the reviewers, 10.1590/0074-02760160334). Statement now reads: “. In contrast, BNZ shows better efficacy clinically when treatment is initiated earlier following infection⁸³ or in asymptomatic patients^{79,80}.” (see lines 802-803 in the clean manuscript version (868-870 in the track-changes version)) and “As the BENEFIT clinical trial demonstrated⁵, BNZ treatment is insufficient to prevent adverse clinical outcomes in chronic symptomatic CD patients, including those with stage III heart failure. In contrast, greater improvement is observed if treatment is initiated in asymptomatic CD^{79,80}” (lines 516-517 in the clean manuscript (555-558 in the track-changes version)). While immunosuppression was not possible in this study (because it would have strongly altered cardiac metabolism), in a separate study we have shown that mice with sterile cure (confirmed by immunosuppression) still present with persistent urine metabolic alterations, supporting the idea that the persistent metabolic alterations following benznidazole treatment in our system are not due to parasite persistence in tissues. We have added this to the discussion, (lines 875-885 in the clean manuscript version (lines 947-957 in the track-changes version)).

5) The main reason for failure of benznidazole treatment to change survival in patients with late stage chronic Chagas disease cardiac form is the extensive areas of transmural scar and LV remodeling. Viable myocardium tissue is replaced for scar creating reentrant circuits for life threatening arrhythmias and inducing LV remodeling with overt heart failure. After this stage, the elimination of the parasite will not restore the heart to its normal size and function. The authors did not discuss these aspects among the reasons for treatment failure in patients with chronic Chagas disease cardiac form.

Response: We agree that cardiac fibrosis and scar tissue are indeed major contributors to the functional impairments observed in chronic Chagas disease and have added this to the text (lines 518-520 and 798-799 in the clean manuscript version (lines 560-562 and 864-867 in the track-changes version)).

6) Please correct: “CD is endemic in Latin America but has become a global health issue due to immigration” for “CD is endemic in Latin America but has become a global health issue due to migration”.

Response: We have made the requested change. Sentence now reads: “CD is endemic in the Americas but has become a global health issue due to migration.” (line 68 in the clean manuscript (line 75 in the track-changes version)).

7) Please, the criteria for the indeterminate form diagnosis is a normal ECG and normal x ray contrast studies of the esophagus and large bowel. Patients with the indeterminate or at the

initial stages of the cardiac form can be both asymptomatic. Please, correct your sentence: “The chronic phase has four forms: the indeterminate form with no apparent clinical symptoms, ...”

Response: As recommended, we have rephrased this sentence. Sentence now reads: “The chronic phase has four forms: an asymptomatic form with no apparent clinical symptoms, a cardiac form, a digestive form, and a cardiodigestive form.” (lines 70-72 in the clean manuscript (lines 76-78 in the track-changes version)).

8) Please, do not use the expression “Chagasic patients”. The association of the people with Chagas disease (Home EN – FINDECHAGAS) does not call themselves this way. This expression should be avoided.

Response: We thank the reviewer for this suggestion and now use people with CD instead (line 94 in the clean manuscript (line 101 in the track-changes version)).

9) The study collaborators handling mice treatment could not be blinded. However, all other experiments could have been done by collaborators blinded to the treatment arm the animals were included. Please, further clarify blinding in this study and discuss this issue in limitations in case blinding was not enough.

Response: We apologize for our lack of clarity. Extractions were performed in batches delineated by each heart section and timepoint. Within a given extraction batch, samples were blinded to treatment group. During LC-MS data acquisition, samples were not blinded, but were fully randomized within a given sampling site, to prevent batch effects. We have added these details in the Methods section, lines 233-234 and 274 in the clean manuscript version (lines 246-248 and 289 in the track-changes version). Thus, metabolomics data acquisition was not influenced by treatment arm.

10) The use of isoflurane to anesthetize mice cause cardiac depression with lower heart rate and LV fractional shortening. However, tables (1, 2 and 3) provided by the authors do not display any results. Only color codes. It is important to display the mean values of the studied parameters for the readers identify their changes with treatment.

Response: All groups were treated the same, so the use of isoflurane would not affect inter-group comparison. We used a standard approach for echocardiography, using isoflurane inhalation anesthesia between 1-3%, and core body temperature between 36.5C to 37.5C to minimize anesthetic and hypothermia effects on cardiac function (Lindsey et al, 2018). Nevertheless, we have provided the median and interquartile range values in Supplementary table 6 (supporting Table 1), and mean values in Supplementary tables S2 and S5.

11) Authors decided to use parametric or non-parametric tests based of their assumption of data normality. However, there are tests that should be used to assess the normal distribution of the data. Thus, please apply the appropriate tests.

Response: We thank the reviewer for this suggestion and have performed Shapiro-Wilk normality tests for representatives of each data type. In all cases, results indicated non-

normality, and thus non-parametric tests have been implemented throughout. We have added this information to the methods, lines 181, 265 and 317 in the clean manuscript version (lines 192, 279 and 332 in the track-changes version).

12) Different cardiac regions have differences in the myocyte structure and myofibers types. Please discuss these differences in the light of your results.

Response: We have expanded our discussion to reflect literature on these differences (lines 773-782 in the clean manuscript version (lines 839-847 in the track-changes version)).

13) Heart septum was included in the left ventricle specimens?

Response: Only left ventricle free wall, not including heart septum, was analyzed. We have clarified this in Methods, line 177 (188 in the track-changes version).

14) “We then sought to determine whether specific metabolites that failed to be restored by treatment could be associated with persistent disease symptoms.” The authors did not test disease symptoms. They tested associations with findings of the echocardiogram and ECG of their mice model. Please, rephrase.

Response: As recommended, we have rephrased to clearly specify the correlations we are performing. Sentence now reads: “We then sought to determine whether specific small molecules that failed to be restored by treatment could be associated with persistent immune alterations or functional cardiac readouts.” (lines 685-687 in the clean track-changes version (lines 742-744 in the track-changes version)).

15) Discussion: “...and the low parasite burden sometimes disconnected from lesion sites in human CD patients 56,63-6”. This can be discussed. The finding of few parasites colonies in a sample tissue of a patient with chronic infection does not mean that during the long term course of the disease, the parasite burden was always low in the current damaged tissue.

Response: We agree with the reviewer that endpoint measurements do not reflect prior parasite distribution dynamics and have added this to the discussion (lines 790-792 in the clean manuscript version (lines 855-857 in the track-changes version)).

16) Conclusions: “ Overall, these results have major implications for CD drug development, providing a mechanism to explain prior clinical treatment failures as well as a path to assess the superiority of novel treatment regimens in pre-clinical animal models. Importantly, these results and our prior work during acute-stage treatment and in vitro infection 33,62 confirm a spatial disconnect between clinical outcomes and parasite burden that should inform CD drug development.” The data obtained from a clinical model in small mammals is useful to generate hypothesis for the pathophysiology of human Chagas disease. There is a long distance between an infection model and patients with decades lasting chronic Chagas disease. Therefore, the authors conclusion that their findings explain the failure of clinical treatment in humans should be toned down. The same apply for the Abstract.

Response: We agree with the reviewer and have emphasized throughout that our data provide a hypothesis for causes of treatment failure in humans, rather than a definitive mechanism and have toned down our phrasing accordingly (lines 39, 122-125 and 948-950 (lines 43, 131-134 and 1021-1024 in the track-changes version). We further highlight how additional studies in humans should be performed to test this hypothesis (lines 800-802 in the clean manuscript version (lines 867-868 in the track-changes version)).

17) There is no survival data comparing the two arms treatment. The main goal to evaluate the success of any treatment in patients with chronic Chagas disease cardiac form is not parasite clearance, but improved survival and quality of life.

Response: No deaths occurred in any group during the course of this experiment. We wish to note that this is not an acutely lethal model, and all endpoints were well before the end of the mouse lifespan. Thus, similar to most chronic Chagas disease mouse models, animal survival was not an efficacy readout. Furthermore, we could not let this infection progress to natural animal death, since we needed to obtain comparable samples across all experimental groups for metabolomics. Instead, our analysis of metabolism reflects a novel readout of treatment success that may be better reflective of long-term cardiac health and survival. We have added this to the discussion, lines 842-848 in the clean manuscript version (lines 913-919 in the track-changes version).

Comments from Reviewer #3:

This manuscript follows on from some great previous work that has brought attention to the importance of spatial variability in metabolic dysregulation within organs affected by *T. cruzi* infection. Here the authors have applied their approach to study the impact of the front line anti-parasitic drug benznidazole (BNZ) and an experimental low dose BNZ + therapeutic vaccination combination treatment. The design of the experiment is relatively robust, using parameters for infection, treatment and downstream analysis that are comparable with common in vivo models used in this field, however, there are some questions about controls and replication.

Response: We thank the reviewers for their appreciation of our work. We have addressed all the reviewer's comments (see details below).

The authors identify correlations that reveal relationships between pathogen load, immune responses, cardiac function and metabolism. The results show this is surprisingly complex, for example, cure of infection does not necessarily lead to full normalization of heart function and metabolism. In fact, on some measures the mice do 'better' even though the treatment was inferior in terms of parasite suppression. The data on the variability between the heart's anatomical sub-regions and on the correlation between overall metabolic perturbation and disease severity metadata are both interesting and quite convincing.

Response: We thank the reviewers for their appreciation of our work.

The focus on post-treatment functional outcomes and placing them in the context of immunological and metabolic restoration, rather than just efficacy of drugs against pathogens, is

very welcome, having the potential to advance the field in a new direction. The findings could find clinical relevance in terms of defining candidate components of chronic Chagas heart disease that are reversible and those which are not.

Response: We thank the reviewers for their appreciation of our work.

The main limitations are:

1) Approach to assessing impact of treatment on infection. This is limited to a parasite-specific qPCR assay to detect *T. cruzi* DNA in heart tissue. While the assay is highly sensitive, it is not very informative about whether complete clearance of the infection has been achieved. This is because *T. cruzi* typically infects several other tissues besides the heart. This raises the possibility that systemic inflammation driven by infection of sites outside the heart is influencing the data, undermining the claim of relevance to broader “post-infectious” conditions. It might be feasible to better infer the likelihood of complete clearance if the sensitivity of the H1 strain to benznidazole (e.g. the IC50) were known to be similar to other strains where the same regimen has been proved to be curative.

Response: We agree with the reviewer that extra-cardiac parasites could persist and mention this as a limitation in the discussion (lines 872-875 in the clean manuscript (lines 944-946 in the track-changes version). We have also toned down our mention of post-infectious settings in the title, abstract and text (lines 42-44 and 125 in the clean manuscript (lines 46-49, 135 in the track-changes version)). New title is: “Localized cardiac small molecule trajectories and persistent chemical sequelae in experimental Chagas disease”.

In this study, we were unable to perform immunosuppression to confirm complete parasite clearance, as it would have strongly altered the immune and metabolic profile. However, in a parallel study, we performed metabolic characterization of urine samples, and were therefore able to follow urine analysis with immunosuppression. Even in animals with no detectable parasites after three rounds of cyclophosphamide immunosuppression, considered sterile cure, metabolism still fails to be restored (doi: 10.1101/2023.06.03.543565). In combination with this current manuscript, these results jointly demonstrate that metabolic alterations that persist post-benznidazole treatment are not driven by extra-cardiac parasites, but rather by a post-parasite (but parasite-initiated) failure of metabolism to renormalize. We were able to complete and submit the urine analysis manuscript recently, and now cite this in the discussion (lines 875-879 (lines 947-950 in the track-changes version)). In further support of these persistent metabolic alterations being independent of persisting parasites, we observed better metabolic restoration in mice treated with the combination benznidazole plus vaccine treatment regimen, even though endpoint parasite burden was greater (see discussion, lines 879-885 (lines 950-957 in the track-changes result)).

2) In my view, the authors often exaggerate the similarity of their model to the clinical situation, especially the BENEFIT clinical trial outcomes. Chagas heart disease pathology is multi-faceted and highly variable in its presentation between individuals. However, the authors over-emphasize one specific aspect, ventricular apical aneurysms, which fits their data of stronger

metabolic perturbation localizing to the lower parts of the ventricles. BENEFIT recruited patients with advanced heart disease and the failure of benznidazole to outperform placebo (after 5 years follow-up) was attributed to pathology having reached an irreversible stage. It seems doubtful that treating mice at 69 days post-infection with a 55 day follow-up is comparable to the extent that is implied. Of note, the authors data (Table 1) shows that BNZ treatment reverses most parameters of cardiac dysfunction, which is not concordant with the BENEFIT trial outcome. The data definitely have potential translational value for human disease, perhaps closer to the early chronic asymptomatic/indeterminate stage, but to claim the study provides a mechanism that explains the failure of BNZ in advanced cardiomyopathy patients appears unjustified.

Response: We have toned down our statements throughout the manuscript. See for example lines 39, 122-125, 798-802 and 948-950 in the clean manuscript (lines 43, 131-134, 864-867 and 1021-1024 in the track-changes version).

3) Lines 36, 99, 433, 643 claim that the data provide a mechanism to explain failures of benznidazole in clinical trials. As set out above, I think this goes substantially beyond the limits of what the data actually show.

Response: As clarified above, we have toned down our statements throughout the manuscript in response to the reviewer's comment. See for example lines 39, 122-125, 798-802 and 948-950 in the clean manuscript (lines 43, 131-134, 864-867 and 1021-1024 in the track-changes version).

4) Line 63, due to immigration – suggest “due to migration”, one region's immigrant is another's emigrant after all.

Response: We thank the reviewer for this suggestion and have made the requested change. Sentence now reads: “CD is endemic in the Americas but has become a global health issue due to migration.” (lines 68 in the clean manuscript (line 75 in the track-changes version)).

5) Methods - Amounts of antibodies used in the flow cytometry should be stated.

Response: The amount of antibodies used to stain cells for flow cytometry has been added to the materials and methods, lines 215-222 in the clean manuscript version (lines 227-234 in the track-changes version).

6) Methods - Details of controls used to set gating coordinates for intracellular cytokine staining should be provided.

Response: The details of controls used to set gating coordinates for intracellular cytokine staining has been added to the materials and methods, lines 225-227 in the clean manuscript (236-240 in the track-changes version).

7) Methods – The experimental design means it is not possible to differentiate the effect of the parasite antigen and the adjuvant (TLR4 stimulation) component of the vaccine. It's also not

possible to know whether the low dose BNZ alone or the therapeutic vaccine alone would have reproduced the observed differences in immune, metabolic and cardiac function parameters.

Response: We agree with the reviewer that the current study design does not allow us to differentiate between the effects of the different components of the combination treatment. Nevertheless, the experimental design is sufficient to support our main arguments, 1) that parasite clearance alone is insufficient to fully restore cardiac metabolism (supported by comparisons between BNZ-only, infected and naive), and 2) that immunomodulating treatment regimens, even if unable to fully clear parasites, provide superior metabolic restoration (supported by comparisons between all four groups). These specific conclusions do not require low-dose BNZ only, adjuvant-only or vaccine only, since we make no specific claims as to whether the benefit derives from the adjuvant directly, or other constituents of the treatment regimen. Furthermore, this study builds on prior work by the investigators, in which they'd previously tested BNZ alone, vaccine alone and adjuvant alone, with regards to functional and immunological parameters (PMC9947347). We have expanded our discussion to clarify this, lines 849-861 (lines 920-932 in the track-changes version)

8) Methods - The data appear to come from a single experiment, raising questions about reproducibility.

Response: We have already generated extensive data on the improved treatment efficacy of vaccine-linked chemotherapy in mouse models of acute infection, demonstrating significant reductions in tissue parasite burdens as well as cardiac inflammation and fibrosis (PMC5865041, PMC7865072). Further, we and others have demonstrated the benefits of vaccine-linked chemotherapy on improving cardiac pathology, structure and function in multiple mouse models of infection (PMC9947347; PMC9499242; PMID: 35101415). Therefore, the biological impact of vaccine-linked chemotherapy is reproducible. We have expanded our discussion of these prior studies at lines 831-840 (902-911 in the track-changes version).

We have also now added extensive additional experiments validating our findings in independent infection cohorts. Specifically, we have added supplementary data from one additional independent mouse cohort, confirming the critical results reported in this paper, namely a lack of cardiac metabolic restoration in benznidazole-treated mice (Supplementary data 2, Figure A; described in text at lines 572-573 in the clean manuscript (lines 618-620 in the track-changes version)). RNA-seq analysis on an independent infection cohort also confirmed the lack of immune restoration by BNZ reported in this manuscript (Supplementary data 2, Figure BC; described in text at lines 539-540 (lines 583-585 in the track-changes version)). Furthermore, focusing on RNA-seq expression modules associated with metabolism demonstrated that almost all the modules differing between BNZ-treated vs uninfected mice are also differential between vehicle and uninfected. This is consistent with the metabolomics data indicating that there are initial effects of infection that are not restored by BNZ treatment (Supplementary data 2, Figure D; described in text at lines 573-577 (lines 620-624 in the track-changes version)). RNA-seq module analysis also supported persistent alterations in purine metabolism even after BNZ treatment, and restoration of lipid metabolism changes (Supplementary data 2, Figure D). Increased purine degradation modules in infected and

infected, BNZ-treated mice, is congruent with the lower levels of purines observed in infected untreated and infected treated samples, compared to naive animals (Supplementary data 2, Figure D; described in text at lines 467-469 (509-510 in the track-changes version)). Increased acylcarnitines in our metabolomics data with infection (lines 501-503 (542-544 in the track-changes version), Figure 3) is consistent with decreased beta-oxidation associated transcripts (Table S4).

Lastly, we have also demonstrated in a third independent infection cohort that sterile cure with benznidazole is insufficient to restore the urine metabolome, confirming our results of persistent metabolic alterations that are independent of parasite burden (doi:

10.1101/2023.06.03.543565). This latter finding was in BALB/c mice infected by luciferase-expressing CL Brener parasites, from Discrete Typing Unit (DTU) TcVI (compared to TcI for strain H1 used in this study), thus showing broad applicability of our findings across parasite DTUs.

9) Line 291-297: “Given the specific localization of CD lesions...” and “metabolic trajectories which correlated with sites of clinical CD pathology” This needs further explanation – what do the authors mean by specific localization – to the heart per se, to the subregions, do lesions refer to infected areas, muscle tissue remodelling, aneurysms, sites of nervous system damage etc? Also, perhaps “corresponds with sites” would be more accurate than correlated with sites, since there is no quantification of pathology?

Response: We apologize for our lack of clarity and have rephrased these sentences to clearly state that we are referring to cardiac apical aneurysms. The first sentence now reads: “Given the characteristic localization of CD apical aneurysms” (line 397 in the clean version (line 414 in the track-changes version)). The second sentence has been removed in response to another reviewer comment, with additional details at the end of the paragraph (“Strikingly, the magnitude of small molecule perturbation was highest in the bottom segments of both the left and right ventricles at our chronic 142 days post-infection timepoint, matching with the fact that CD patients commonly present cardiac apical aneurysms, in addition to left ventricle apical and posterior fibrosis”, lines 427-430 in the clean manuscript version (lines 447-450 in the track-changes version)).

10) Figure 1C, is there a way for the p values quoted in the main text (from line 299 onward) to be shown on this figure?

Response: As recommended, we have added stars indicating significant differences between naive and infected groups at each time point.

11) Supplementary File 1, figure legends need more detail – what are the data points, metabolites?

Response: We apologize for our lack of clarity. Each data point is the Bray-Curtis distance in terms of overall metabolite composition between every pair of samples. For example, in the boxplot marked naive_vs_infected_50DPI, this would be the distance between every naive

sample and every infected sample, at 50 DPI, in all possible permutations. We have now clarified this in the figure legends throughout Supplementary Data 1.

12) Line 317: “Strikingly, the magnitude of metabolic perturbation was highest in the bottom segments of both the left and right ventricles ... matching with the sites of CD damage in patients” Similar to above, what type of damage are they referring to? Apical aneurysm? The cited review does not support the idea that Chagas heart disease pathology is localized only to these sub-regions of the ventricles e.g. there can be diffuse inflammation, microvascular damage and denervation in atrial regions.

Response: We apologize for our lack of clarity and have rephrased this sentence to be more specific, as well as added an additional reference covering damage location (Nunes et al, Circulation; PMID 30354432). Sentence now reads: “Strikingly, the magnitude of small molecule perturbation was highest in the bottom segments of both the left and right ventricles at our chronic 142 days post-infection timepoint, matching with the fact that CD patients commonly present cardiac apical aneurysms, in addition to left ventricle apical and posterior fibrosis”, lines 427-430 in the clean manuscript version (lines 447-450 in the track-changes version).

13) Line 378, “even though parasite burden in BNZ-treated animals was no longer significantly different from uninfected animals at all ventricle sites”. I’m not sure making this statistical comparison is valid – uninfected animals by definition have zero parasite burden. Better to say something like the amount of parasite DNA was not significantly higher than the detection limit/threshold? Moreover, the data can’t distinguish between an animal that is completely cured of infection vs one where the amount of parasite DNA in the tissue sample is below the detection limit of the qPCR assay or where the animal is infected but there happened to be no parasites (or parasite DNA) in the piece of tissue used for DNA extraction. There can be relatively high numbers of parasites in a range of tissues e.g. skeletal muscle, gut, even when infection is not detectable in the heart. For this reason, it is important to clarify that parasite clearance (Line 431) only means clearance from the heart. This is acknowledged in the discussion (section at line 608), but what is not considered is the potential systemic effect on the heart of active infection elsewhere in the body, for example, circulating pro-inflammatory cytokines like TNF-alpha.

Response: We have rephrased the statement at line 378 (now at line 535-537 in the clean manuscript (lines 579-580 in the track-changes version)) as recommended by the reviewer. As recommended, text now reads: “even though parasite burden in BNZ-treated animals was no longer significantly different from the limit of detection at all ventricle sites”. We agree with the reviewer that we could not confirm sterile cure or lack of extra-cardiac parasites here, and indeed reviewer #2 also raised this issue. In this study, we were unable to perform immunosuppression to confirm complete parasite clearance, as it would have strongly altered the immune and metabolic profile. However, in a parallel study, we performed metabolic characterization of urine samples, and were therefore able to follow urine analysis with immunosuppression. Even in animals with no detectable parasites after three rounds of cyclophosphamide immunosuppression, considered sterile cure, metabolism still fails to be restored (doi: 10.1101/2023.06.03.543565). In combination with this current manuscript, these

results jointly demonstrate that metabolic alterations that persist post-benznidazole treatment are not driven by extra-cardiac parasites, but rather by a parasite-independent failure of metabolism to renormalize, which could indeed be driven by persistent systemic immune responses or inflammation, as suggested by the reviewer. We were able to complete and submit the urine analysis manuscript recently, and now cite this in the discussion (lines 875-879 (lines 947-950 in the track-changes version)). We have also edited line 431 (now lines 613-616 in the clean manuscript version (lines 661-664 in the track-changes version)) as recommended, which now reads: “These findings also indicate that parasite clearance from the heart is insufficient on its own to restore the small molecule profile to a pre-infection state, and that the inability of BNZ to fully restore the small molecule profile is not due to lingering parasite fragments.”

14) How bad is the heart disease in this model? Table 1 includes parameters that are restored or not restored, but it is hard to tell if this is recovery from a severe disease state or a mild one. Showing the data comparing these parameters for naïve vs infected untreated groups would help the reader to gauge this. It would also improve the ability to compare the model with human disease states.

Response: As recommended, we have added a table covering the raw data supporting Table 1 (Supplementary Table S6). In this infection model, there are already significant changes in cardiac structure and function by about 70 days of infection on echocardiography, suggesting mild to moderate disease (PMC9947347). We consider this model to be well representative of the slowly progressive disease seen in humans that can develop over decades. We have added additional discussion of our mouse model in Methods (lines 148-156 in the clean manuscript version (lines 159-166 in the track-changes version)), and its comparison to human infection in the Discussion (lines 903-928 of the clean manuscript (lines 976-1001 in the track-changes version)).

15) Table 1 – The tissue corresponding to the T cell subsets should be stated – spleen? The data are referred to as %, but this means a % of what? It is unclear whether these data reflect an absolute difference in numbers of each CD8 T cell phenotype or if the %s are the result of shifts in the proportions of cells with other phenotypes.

Response: We now clarify in table 1 that these T cells are measured from splenocytes and are reported as percent of the parent, as described in the gating strategy figure (Supplemental figure S1) and in Methods (lines 229-230 in the clean manuscript (242-243 in the track changes version)).

16) Also for Table 1, what is the justification for using p-value cut offs of 0.1 rather than standard 0.05?

Response: As performed here, cutoffs for FDR-adjusted p-values are often set to 0.1 rather than 0.05, to recognize the effect of the FDR correction (see for example by other authors: PMC8863577 in Nature Medicine, PMID: 30382244 in Nature Protocols).

17) It is interesting that the combo treatment restored the splenic CD8 T cell compartment % to a naïve-like level, but parasite DNA was still present and all cardiac functional parameters were either not restored or scored as unclear impact. Conversely, the full BNZ regimen failed to normalize the CD8 T cells but most cardiac functional parameters were restored. Could this partially be explained by detection of a memory T cell response in the spleen after full BNZ treatment, a response that is absent from the combo regimen group?

Response: We thank the reviewer for their insightful interpretation of these results. Several studies in humans and mouse models have described the impact of curative BNZ treatment on parasite specific memory CD8+ T cell responses (PMID: 14976609; PMCID: PMC8600036; PMCID: PMC3074975). Additionally, a recent study by Dzul-Huchim and colleagues showed that their vaccine-linked chemotherapy strategy did increase memory CD8+ T cell populations (PMCID: PMC9499242). However, for all of these studies, evaluation of the immune responses after treatment were at much later time points after treatment completion than in our study. We did not specifically evaluate memory CD8+ T cell populations in this study, but we acknowledge the importance of evaluating these populations in future studies and evaluating later time points to fully understand the correlations between immune responses, metabolism and cardiac function. We include a discussion of this in lines 834-842 in the clean manuscript version (lines 905-913 in the track-changes version).

18) Figure 3F – it looks like some data points have been cut off as a result of segmenting the data axis

Response: We apologize for this formatting issue and have fixed the axis so that no data is hidden. Figure and figure legend have been updated accordingly (now Figure 4F).

19) Line 460, Table 2 title, “restored by treatment” – which treatment?

Response: We apologize for our lack of clarity. We had meant “restored by any treatment” (by BNZ-only treatment or combo treatment or both). We have now clarified the title, which now reads: “Table 2. Proportion of each small molecule superclass restored by any treatment, per heart segment.” (lines 669-670 in the clean manuscript (lines 725-726 in the track-changes version)).

20) Line 497 to 525, I think this section would benefit from some specific example metabolites that come out of the analysis as candidates for being linked to persistence of heart disease symptoms. Perhaps those with the highest correlation co-efficients?

Response: We have added examples as recommended (lines 717-721 in the clean manuscript (lines 775-779 in the track-changes version)).

21) I find it hard to get a sense of how large the overall metabolic dysregulation is relative to the total metabolome. For example, in Figure 4 the charts encompass up to ~250 metabolites (the large majority of which are classified as “no matches”) but what kind of fraction is 250 of the total number of metabolites that were not significantly different between any of the groups?

Response: As requested, we have calculated the proportion of differential metabolite features, compared to total detected metabolite features. For time-course analysis, between 0.33% of detected metabolite features (in right atrium) and 5.14% of detected metabolite features (in right ventricle top) were affected at at least one timepoint, with comparable proportions also affected in the left ventricle bottom (4.43 %) and right ventricle bottom (3.46%). Small molecules impacted by treatments represent 0.87% to 5.54% of detected features, depending on the sampling site. We have added this information to the text (lines 456-459 and 624 in the clean version (lines 494-497 and 675-676 in the track-changes version)).

22) Line 590, purinergic neurotransmission in the heart may be relevant to the discussion of purines.

Response: We thank the reviewer for this suggestion and have expanded our discussion of the relevance of this result to purinergic neurotransmission (lines 811-827 in the clean manuscript version (lines 878-896 in the track-changes version)).

Comments from Reviewer #4:

I read with interest this article on the treatment of a neglected tropical disease (Chagas) caused by *T. cruzi* in an animal model. The authors applied a typical antiparasitic (BNZ) and tested whether an immunomodulator (Tc24-C4), would ameliorate the long term cardiac effects of the disease. This is an important question and new drug treatments are vitally needed in order to combat what is likely to be an increasingly prevalent disease. While I have some considerable misgivings regarding the data collection methodology used, the biology appears to bear out that the combination therapy has a significant impact on long term disease prognosis.

Response: We thank the reviewer for their appreciation of the importance of our findings, especially in the context of Chagas disease treatment.

1) The authors justify a metabolomics approach to this study by stating that there is a correlation of metabolite levels to inflammation and cytokine levels, and that these are therefore causally linked. Since this is the entire premise of the paper, a stronger link should be more clearly delineated, since it is not very obvious how fibrosis and overall heart inflammation is linked to metabolomics, especially the panel of metabolites detected in this study.

Response: We thank the reviewer for this suggestion and have accordingly expanded our discussion of the link between small molecules (including lipids), inflammation, cytokines and fibrosis in the introduction (lines 98-115 in the clean manuscript version (lines 106-123 in the track-changes version) and in the discussion, lines 888-896 in the clean manuscript (lines 960-968 in the track-changes version). In response to this reviewer's suggestion, below, we have also generated a summary figure to better integrate these concepts (Fig. 9).

2) I have some concerns about the metabolomics - while sample preparation from tissue was managed very well, the sample preparation is clearly geared towards polar and non-polar

extractions. These fractions are then resuspended using the same solvent and injected on a C8 column. There are several missing details - what was the flow rate of the column?

Response: We apologize for the confusion. The flow rate was 0.5 mL/min. This information can be found in Table S1.

3) This is important as injecting samples in a high strength (50% MeOH) buffer can have impacts on the retention time of some compounds. I am also somewhat concerned about the breadth of compounds resuspended from each fraction in the chosen solvent. The combination of a weak reversed phase column and the analytical methodologies chosen have led to what I would describe as a small number (84) of annotated metabolites, nearly all of which are lipids (73).

Response: We apologize for not better describing the rationale for our workflow. Additional details have been added throughout the manuscript (lines 234-236 and 271 (246-249 and 285 in the track-changes version)). We used a method for small molecule extraction and resuspension published by Want et al in Nature Protocols (PMID: 23222455) that has been cited over 600 times. They specifically recommend the two-step extraction procedure that we use, as well as resuspension in 50% methanol of both polar and non-polar fractions prior to data acquisition. This is now clarified at lines 234-236 and 271 (246-249 and 285 in the track-changes version). We have previously validated that this specific extraction method, resuspension procedure, LC separation and MS data acquisition workflow shows a proportional relationship between multiple polar and less-polar small molecules vs peak area in the samples, that match the chemical classes discussed in this manuscript, including nucleobases, amino acid derivatives and lipids (PMC9426520) (also specified at lines 301-304 (lines 316-319 in the track-changes version)).

We used a strict cutoff for annotation: all reported annotations are supported by MS2 spectra, rather than only relying on *m/z* database matching. All MS2 matches were visually inspected and filtered for biological plausibility, as recommended by PMC9979140. We have expanded our Methods section to clarify this (lines 343-347 in the clean manuscript version (lines 359-363 in the track-changes version)). Our annotation rate in Table S2 is 24.2% (84/347) and in Table S3, 20.7% (40/203). This is no lower than what is commonly reported for molecule analyses using MS2 matching as an additional filter beyond *m/z* (see for example PMC9722809, PMID: 35798960). We now clarify this in the text at lines 347-349 (clean version; lines 363-365 in the track-changes version).

Furthermore, we wish to point out that the manuscript tables only describe small molecules impacted by infection, which indeed appear to be mainly lipids as well as nucleotides. However, overall detected small molecules include broader structural properties. Analysis of all detected features using Classyfire revealed molecules belonging to multiple glycerophospholipid subclasses, fatty acid esters, amino acids, peptides, amines, carbohydrates, fatty acids, fatty amides, alcohols, purines, and eicosanoid subclasses (lines 349-352 in the clean manuscript version (lines 365-368 in the track-changes version)). Thus, this enrichment of lipids and nucleosides in infection-impacted molecules is more likely a consequence of specific effects of infection rather than of our workflow.

4) It might be better to describe the study as a lipidomics one, rather than a metabolomics one.

Response: We apologize for the confusion. We had used “metabolomic” in its broadest sense of all small molecules <1500 Da (including lipids, as defined by the Human Metabolome Database), and now clarify this (lines 95-98 in the clean manuscript (lines 102-105 in the track-changes version)). To avoid confusion, we now use predominantly the term “small molecules” rather than metabolites in our title (now changed based on reviewer suggestions to “Localized cardiac small molecule trajectories and persistent chemical sequelae in experimental Chagas disease”) and throughout the text. As clarified above, we detect and annotate many more small molecule classes than just lipids, including amino acids and small peptides, nucleotides, and carbohydrates (see lines 349-352 in the clean manuscript version (lines 365-368 in the track-changes version)).

In addition, in accordance with the reviewer’s suggestion, we have also expanded our lipid analysis and discussion, by performing systematic analysis of acylcarnitines and glycerophospholipids. Our rationale for focusing on these lipid classes is that they were the ones that had individual members impacted by infection. This analysis revealed increases in all three lipid classes with infection and a pattern of renormalization of acylcarnitines with benzimidazole treatment, consistent with RNA-seq data that we have added to the manuscript. We have now added figures 3, 5 and Supplementary Data 3 to reflect this, with additional text at lines 486-503 and 579-582 in the clean manuscript (lines 527-544 and 626-629 in the track-changes version).

5) I was also curious about the choice of running only in positive mode. Succinic acid and itaconate, two metabolites associated with inflammation, for example, are detected primarily in negative ionisation mode, and would recommend the choice of a HILIC column or GC-MS rather than a C8 column, where all the polar metabolites are very likely to be found in the wash-through. Was there a reason for the choice of methodology?

Response: We apologize for not better describing the rationale for our chromatography choices and choice of positive mode. While we agree that in principle multiple chromatography conditions and different ionization modes indeed lead to expanded chemical coverage, building spatial maps of small molecules necessitates many samples (6 per mouse in this study, with N=15 mice per group), and it is thus not logistically feasible to perform all possible chromatography and polarity combinations. Instead, we selected methods that we have previously validated (see PMC9426520), and that matched with our prior work on Chagas disease, enabling cross-study comparison of findings and thus greater biological insight. Furthermore, our prior work demonstrates that these conditions are suitable to detect chemical classes that are impacted by *T. cruzi* infection, which we know include multiple lipid classes that ionize well in positive mode (acylcarnitines, glycerophosphocholines). However, as discussed above, our chromatography approach still enables the detection of many polar small molecules, and we have demonstrated that our chromatography and MS method enables quantification of both polar and less-polar small molecules (PMC9426520), across the chemical classes discussed in the manuscript. We clarify this in Methods, lines 289-293 and 349-352 in the clean manuscript version (lines 304-308 and 365-368 in the track-changes version)). In previous work

(PMC7385396), we had performed both positive mode and negative mode acquisition. However, annotation rates were three times greater in positive mode than in negative mode, likely reflecting the greater number of positive mode spectral libraries available. Adding negative mode data acquisition would have doubled our run time and costs, without unfortunately leading to double the number of annotations. This is now specified at lines 296-300 (lines 311-315 in the track-changes version).

6) Additionally, were system equilibration runs performed prior to the batch? System equilibration runs are best practice for untargeted metabolomics, stabilising the system for the rest of the batch and the equilibration is reset by running interspersed blanks, as described in the methods, I am therefore confused as to why this was done. At the bare minimum, a PCA should be included in the supplementary data, allowing the reader (and reviewer) to assess the data quality, clustering of samples, and the reproducibility of the pooled samples.

Response: We did indeed perform system equilibration prior to run start, using 5 pooled QC injections. We apologize for not clearly specifying it in the original manuscript version. It is now specified at lines 273-274 in the clean manuscript (lines 288-289 in the track-changes version).

In a large sample run like this one, we have found that running blanks only at start and end are insufficient to monitor for carryover. Importantly, every blank was followed by a pooled QC, rather than by a sample. We apologize for not clearly specifying this in the original manuscript. We now do so (line 275 in the clean manuscript (line 290 in the track-changes version)). Interspersed blanks have also been critical to diagnose autosampler malfunction, situations where the autosampler picks up from a well different from the one indicated in the run sequence. Though rare, such an issue if undiagnosed would lead to critical data misinterpretation. In the single case where this happened in our laboratory, we were able to notice and correct the issue by realizing that samples marked as blanks didn't look "blank", and then watching the autosampler movement carefully, followed by instrument resetting. In a more routine run, inclusion of blanks at regular intervals enables the monitoring of carryover, so that any "sticky" small molecules can be filtered out.

Using a single blank followed by one sample (in this case, one QC) is reported in the literature to have only a mild effect, leading to significant changes in retention time and peak width in fewer than ~20 features in subsequent samples (PMC6614502). Magnitude of impact is also important - again, examples of affected features in the literature indicate retention time shifts following blanks of no more than 0.025 min, with little to no discernable impact on peak area in presented examples (PMC6614502). Such small retention time shifts are readily addressed by our feature alignment parameters, and are very minor compared to peak width. Furthermore, because all samples were run in randomized order, this would have no systematic effect on data that could bias interpretation. We now clarify this at lines 275-277 in the clean manuscript (lines 290-292 in the track-changes version).

As recommended, we have added a PCoA showing blanks and QCs, Figure S6. Note that the clustering of the QCs apart from the samples is expected, since they used different injection volumes (5 μ L for pooled QC vs 30 μ L for samples). This difference was necessary to ensure

that we had enough QC material to inject throughout the run. We could not prepare larger volumes of pooled QC due to the limited available material in some smaller sections like the atria. Overall, biological variation exceeded technical variation. Further evidence of the robustness of our analyses is provided by the overlapping conclusions between our small molecule analysis results and our RNA-seq results (Supplementary Data 2)

Lastly, to address the reviewer's concern and to further confirm that the inclusion of mid-run blanks did not cause appreciable mid-run deconditioning, we compared chromatograms from samples immediately preceding the blank and samples immediately following the blank and pooled QC, using four representative samples from the right ventricle top. Total ion chromatograms showed excellent overlap in terms of retention times. Differences in some peak intensities in the total ion chromatogram were observed, but are to be expected since these are different experimental samples, from different experimental groups. This is now Figure S2. Comparing retention time and peak area of the extraction control (sulfachloropyridazine) likewise did not demonstrate any systematic effect of the mid-run blank + QC on the subsequent sample, with excellent alignment between the pre-blank and post-blank samples. This is provided as Figure S3.

7) I struggled to understand the PERMANOVA/pseudo-F trajectories figure (1 C). What do the trajectories actually show and what are the criteria for change?

Response: As recommended, we have clarified the interpretation of Figure 1C in the text (line 407 in the clean manuscript (line 426 in the track-changes version)) and in the figure legend (lines 446-450 in the clean manuscript (lines 481-488 in the track-changes version)). A greater pseudo-F value represents a greater effect size: a greater difference between infected and uninfected samples at that site and timepoint. The effect size values are complemented by PERMANOVA p-values, with a significance cutoff of $p < 0.05$. Per recommendation of reviewer 3, we have added significance stars to the figure, which we hope will further assist with interpretation.

8) I would expect to see more mechanistics overall in the discussion - a figure explaining how this all works would be very helpful - why and how are these metabolites specifically changing, and how is this linked to the intrinsic biology that is going on in the tissue? There are several examples of individual metabolites changing, but little (apart from 'positive' or 'negative' correlation) linking them with the process of inflammation and fibrosis that presumably the immunomodulator is intended to remediate.

Response: As recommended by the reviewer, we have expanded our discussion of the link between infection, immunity and the observed small molecule changes (lines 811-816 and 888-896 (lines 879-883 and 960-968 in the track-changes version)). As recommended by the reviewer, we have also added a conceptual figure with a proposed model of the intersection between parasite and immune effects on metabolism pre- and post-treatment (Figure 9).

9) Finally, normalisation is a knotty problem in tissue metabolomics, especially instances like Chagas where fibrosis changes the composition of the tissues. In the methods section,

normalisation is performed at the sample level via wet weight (50mg), and then by TIC - what was the impact of TIC normalisation, and was this a straight numerical TIC modifier, or was something more complex like quantile normalisation used?

Response: We apologize for our lack of details. Data was normalized to constant sum of 1 for each sample. We have added this to the methods, line 314 in the clean manuscript (line 329 in the track-changes version). TIC normalization vs no normalization had very minor effects on the data, as evidenced by comparable PCoA plots, with comparable % variance explained by each axis (Supplemental Figure S6). Normalization did slightly improve clustering with regards to sampling site, leading to clearer clustering of atrial vs ventricular samples. Consequently, we used TIC-normalized data throughout the manuscript.

10) In all, while the principles of the work are very good, the metabolomics data is not ideal to support the conclusions.

Response: In response to the reviewer's comments and as detailed above, we have extensively expanded the justification of our methods, highlighting how they have been thoroughly validated, and are based on a Nature Protocol article extensively cited (PMID: 23222455). We further highlight how these methods are consistent with our prior work on Chagas disease, enabling cross-study comparisons and detection of molecule classes expected to be impacted by infection and by treatment. We also highlight how these methods have been validated to enable comparisons of polar and non-polar small molecules between experimental groups (PMC9426520). Given that the term "metabolite" can be easily misconstrued (either to refer to central metabolism, or instead to broadly include all small molecules less than 1500 Da, including lipids), we now instead privilege the term "small molecules" throughout the manuscript. Our methods are thus well appropriate to study the impact of infection and treatment on multiple small molecule classes. We have also expanded our lipidomic data analysis, in accordance with the reviewers' comments. A particular finding is an increase in acylcarnitines with infection, and a pattern of restoration of this effect with benznidazole treatment, which is further consistent with additional RNA-seq data which we have added to the manuscript. We have also added additional data supporting impaired small molecule and metabolic restoration after benznidazole treatment in replicate cohorts (Supplementary Data 2). Our LC-MS results also concur with RNA-seq data showing that BNZ treatment did not restore purine metabolism but restored several pathways associated with lipids, further supporting our approach and the resulting data. This has been added as Supplementary Data 2. We now believe that the manuscript is much strengthened, with our LC-MS data supporting our major conclusions of persistent small molecule changes following benznidazole treatment, that are partially alleviated following combination treatment, as well as specific effects on several small molecule classes, including purines and lipids.

REVIEWER COMMENTS

Reviewer #2 (Remarks to the Author):

Dear Authors,
I have no further comments.

Reviewer #3 (Remarks to the Author):

The ms is substantially improved, the authors have comprehensively addressed all of my comments and questions.

Reviewer #4 (Remarks to the Author):

I thank the authors for responding to my comments. Some points, however, require further clarification:

1): The added text goes some way to justifying the link between inflammation and small molecule changes – no further changes are required.

2, 4, 5, 7, 8, 9 and 10): No further changes.

3): Further clarification is required here:

“Furthermore, we wish to point out that the manuscript tables only describe small molecules impacted by infection, which indeed appear to be mainly lipids as well as nucleotides. However, overall detected small molecules include broader structural properties.”

Can you clarify the above, please? From my analysis of the CSV file provided, there are 347 features detected in the analysis, of which 84 are annotated, 73 of which are lipids. Is this table missing features that are not changing in one or more tissues/time points? If so, what was the total number of features detected? Otherwise, are the Classyfire results based on these 84 annotations, if so, from the 11 non-lipid annotations there will be only a single member in most categories to suggest ‘broader structural properties’. Is this the case?

You also say that all of the annotations were supported by MS2 data – how were these data queried? Via a database search with a scoring scheme, then manual assessment, or purely by manually assessing the patterns? Regardless of the method, what criteria were selected to determine a match/non-match?

6) Please provide the S6: PCA images with PC1/2 in two dimensions. Without the capability to rotate the 3D maps as shown it’s challenging to assess the level of overlap on each 2D plane.

Response to reviewer comments

Reviewer #4

3) Further clarification is required here:

“Furthermore, we wish to point out that the manuscript tables only describe small molecules impacted by infection, which indeed appear to be mainly lipids as well as nucleotides. However, overall detected small molecules include broader structural properties.”

Can you clarify the above, please? From my analysis of the CSV file provided, there are 347 features detected in the analysis, of which 84 are annotated, 73 of which are lipids. Is this table missing features that are not changing in one or more tissues/time points? If so, what was the total number of features detected? Otherwise, are the Classyfire results based on these 84 annotations, if so, from the 11 non-lipid annotations there will be only a single member in most categories to suggest 'broader structural properties'. Is this the case?

Response: Supplementary table 2 only covers features that were impacted by infection, and supplementary table 5 only covers features responding to infection and treatment. They do not represent all of the detected features. Features that are not changing in response to infection are not covered in these tables. We have rephrased our table titles to improve clarity. Supplementary table 2 is now titled: “Small molecule features impacted by infection over time.” and supplementary table 5 is now titled “Small molecule features impacted by treatment.” (lines 1289-1290 and 1293-1294 of the expanded track-changes view; lines 1002 and 1005 in the clean version).

The total number of features detected was 6,712. The statement in methods (now at lines 1205-1208 in the expanded track-changes view (lines 920-923 in the clean manuscript) refers to this full dataset. We have clarified this statement to indicate that it refers to the full dataset, rather than just to infection-impacted or treatment-impacted features. (“Detected Classyfire subclasses **across all detected small molecule features** included fatty acid esters, amino acids, peptides, multiple glycerophospholipid subclasses, amines, carbohydrates, fatty acids, fatty amides, alcohols, purines, and eicosanoids.”). Links to the full annotation workflow and to the MolNetEnhancer workflow that provided the ClassyFire annotations are provided in the Data Availability section, lines 1241-1245 of the expanded track-changes version (lines 956-960 in the clean version) and reproduced here for convenience: “Molecular networks can be accessed at <https://gnps.ucsd.edu/ProteoSAFe/status.jsp?task=dd1ef14c8a964bfd8843da96aa957d89> (feature-based molecular network) and <https://gnps.ucsd.edu/ProteoSAFe/status.jsp?task=3fefbc8549604d34954aba2a95ec79df> (MolNetEnhancer 44)”.

You also say that all of the annotations were supported by MS2 data – how were these data queried? Via a database search with a scoring scheme, then manual assessment, or purely by manually assessing the patterns? Regardless of the method, what criteria were selected to determine a match/non-match?

Response: All our annotations were generated using feature-based molecular networking,

which uses MS2 spectra from the collected data to query the spectral libraries available in the Global Natural Products Social Molecular Networking (GNPS) platform and to build networks of features with similar spectra within the data, with a similarity cutoff of ≥ 0.7 cosine score and ≥ 4 matched peaks. Annotations from feature-based molecular networking are obtained two ways. First, MS2 spectra from the data are matched to the GNPS libraries. Only spectral matches with ≥ 0.7 cosine score and ≥ 4 matched peaks were retained. From this output, we only retained annotations that were within 10 ppm and biologically plausible. We also visually inspected all these spectral matches using mirror plots, to confirm success of the cosine scoring algorithm. Secondly, network annotation propagation can be performed, building on the network structure to expand from these direct library matches to other features with similar MS2 spectra (within the same molecular subnetwork), but allowing for mass differences that reflect common chemical differences. In this case, spectral similarity cutoffs were also ≥ 0.7 cosine score and ≥ 4 matched peaks. These are marked as “annotated through molecular networking” in our supplemental tables. We have provided an expanded explanation of molecular networking in Methods at lines 1187-1202 of the expanded track-changes view (lines 902-917 in the clean version), along with full molecular networking parameters (lines 1190-1193 in the expanded track-changes version (lines 905-908 in the clean version)). The full molecular networking output URL is also provided in the Data Availability statement, lines 1241-1245 of the expanded track-changes version (lines 956-960 in the clean version), so that anyone can repeat and reproduce our analysis.

6) Please provide the S6: PCA images with PC1/2 in two dimensions. Without the capability to rotate the 3D maps as shown it's challenging to assess the level of overlap on each 2D plane.

Response: We have provided two-dimensional PCoA plots as requested (Supplementary Figure 6).

REVIEWERS' COMMENTS

Reviewer #4 (Remarks to the Author):

We thank the authors for their clarifications on the metabolomics and have no further issues with recommending the paper for publication.